# Mapping the functional versatility and fragility of Ras GTPase signaling circuits through in vitro network reconstitution

**Scott M Coyle[1,2,3,4], Wendell A Lim[1,2,3,4]\***

[1]Howard Hughes Medical Institute, University of California, San Francisco, San Francisco, United States; [2]Department of Cellular and Molecular Pharmacology, University of California, San Francisco, San Francisco, United States; [3]Program in Biological Sciences, University of California, San Francisco, San Francisco, United States; [4]Center for Systems and Synthetic Biology, University of California, San Francisco, San Francisco, United States

**\*For correspondence:** Wendell. Lim@ucsf.edu

**Competing interests:** The authors declare that no competing interests exist.

**Abstract** The Ras-superfamily GTPases are central controllers of cell proliferation and morphology. Ras signaling is mediated by a system of interacting molecules: upstream enzymes (GEF/GAP) regulate Ras's ability to recruit multiple competing downstream effectors. We developed a multiplexed, multi-turnover assay for measuring the dynamic signaling behavior of in vitro reconstituted H-Ras signaling systems. By including both upstream regulators and downstream effectors, we can systematically map how different network configurations shape the dynamic system response. The concentration and identity of both upstream and downstream signaling components strongly impacted the timing, duration, shape, and amplitude of effector outputs. The distorted output of oncogenic alleles of Ras was highly dependent on the balance of positive (GAP) and negative (GEF) regulators in the system. We found that different effectors interpreted the same inputs with distinct output dynamics, enabling a Ras system to encode multiple unique temporal outputs in response to a single input. We also found that different Ras-to-GEF positive feedback mechanisms could reshape output dynamics in distinct ways, such as signal amplification or overshoot minimization. Mapping of the space of output behaviors accessible to Ras provides a design manual for programming Ras circuits, and reveals how these systems are readily adapted to produce an array of dynamic signaling behaviors. Nonetheless, this versatility comes with a trade-off of fragility, as there exist numerous paths to altered signaling behaviors that could cause disease.

## Introduction

Many dynamic processes in the cell such as proliferation, differentiation, or morphological change are regulated by signaling through members of the Ras superfamily of small GTPases (*Chang et al., 2003*; *Sjölander et al., 1991*; *Hofer et al., 1994*; *Vojtek and Der, 1998*; *Bourne et al., 1990*). Mutations in these important molecules are often associated with cancer or other diseases (*Bos, 1989*; *Schubbert et al., 2007*). These small GTPases act as macromolecular 'switches' at cell membranes, cycling between an ON state when bound to GTP and an OFF state when bound to GDP (*Bourne et al., 1990*) (*Figure 1A*). This notion of an ON and OFF state of the GTPase is manifest in differences in the conformation of the protein such that, in most cases, only the GTP-bound state is able to interact with downstream effector molecules and assemble signaling complexes (*Krengel et al., 1990*; *Milburn et al., 1990*; *Nassar et al., 1995*; *Herrmann, 2003*).

**eLife digest** Cells sense and respond to the world around them using signaling "circuits" made of proteins and other molecules, and when an important cell circuit breaks, diseases like cancer may arise. Much like with electrical circuits, a given set of molecular components can be used to build different signaling circuits that behave in different ways. However, unlike for electrical circuits we generally do not have design manuals that allow us to work out how a signaling circuit behaves based on the components it includes. Doing this would involve identifying all the molecular parts of a circuit, using them to build every possible circuit, and carefully measuring the associated behavior.

A group of proteins called the Ras-superfamily GTPases are important controllers of cell behavior. To investigate the behavior of Ras GTPase signaling circuits, Coyle and Lim built up different circuits from their components and "watched" their behavior with a microscope. Analyzing these behaviors provided the information needed to produce a 'design manual' for programming Ras circuits.

Coyle and Lim found that the makeup of a Ras signaling circuit strongly affects the timing, duration, shape and size of its output. This means that different cells can use the same core components in different ways to build circuits customized to their specific needs. Nonetheless, this versatility comes with a trade-off: the circuits are fragile, and can break in many different ways to cause disease.

In the future Coyle and Lim aim to build other types of important cellular signaling circuits from their component parts. Only by building these systems, turning them on and watching them run can we begin to understand how they actually perform and what they are capable of.

As enzymes, these GTPases are formally capable of binding GTP, hydrolyzing it to GDP+$P_i$, and releasing product to complete the catalytic cycle on their own, but, in practice, the GTPase is incredibly slow at each stage of this cycle except for the initial binding of nucleotide (*Neal et al., 1988*; *Gibbs et al., 1984*; *McGrath et al., 1984*). As such, molecules that can accelerate these slow steps in the catalytic cycle function as essential regulators of GTPase activity during signaling events: guanine exchange factors (GEFs), which promote product release by emptying the nucleotide pocket of the GTPase and allowing subsequent reloading of the GTPase with nucleotide (OFF->ON transition); and GTPase-activating proteins (GAPs) which accelerate the hydrolysis of GTP to GDP+$P_i$ (ON->OFF transition) (*Figure 1A*) (*Boguski and McCormick, 1993*; *Trahey and McCormick, 1987*; *Bos et al., 2007*; *McCormick et al., 1991*). How Ras processes information, then, is not determined by Ras alone, but rather is also highly dependent on a system of molecules comprising the upstream GEFs and GAPs that regulate its activity, and the downstream effector molecules that are engaged and regulated by the activated GTPase (*Figure 1B*).

Our biochemical view of Ras and Ras-associated proteins, however, is largely focused on individual molecules, rather than the system of molecules. Considerable work in vitro over several decades has provided structural and biochemical insights into how individual GEFs, GAPs and effectors function, as well as how their activity can be controlled through mechanisms like autoinhibition and allostery (*Lenzen et al., 1998*; *Sondermann et al., 2004*; *Margarit et al., 2003*; *Iwig et al., 2013*; *Scheffzek et al., 1997*; *Boriack-Sjodin et al., 1998*; *Feng et al., 2004*; *Bollag and McCormick, 1991*). These studies have provided many of the most fundamental insights into the mechanisms of Ras activation and inactivation as well as clarified our understanding of the nature of oncogenic mutations. To study the regulators, however, these reconstitutions are almost always performed under conditions in which Ras cannot productively cycle: GEF assays monitor a single exchange of fluorescent nucleotide for non-fluorescent nucleotide; GAP assays monitor a single turnover of GTP without the possibility of the nucleotide reloading (*Eberth and Ahmadian, 2001*). Likewise, studies of effector interactions with activated GTPase are typically done under non-cycling conditions using non-hydrolyzable nucleotide analogs to measure equilibrium binding constants (*Geyer, 1996*; *Herrmann et al., 1996*; *Sydor et al., 1998*). However, we know that signaling dynamics are critical for many cellular responses, and yet these features have not been analyzed in most in vitro studies of Ras signaling.

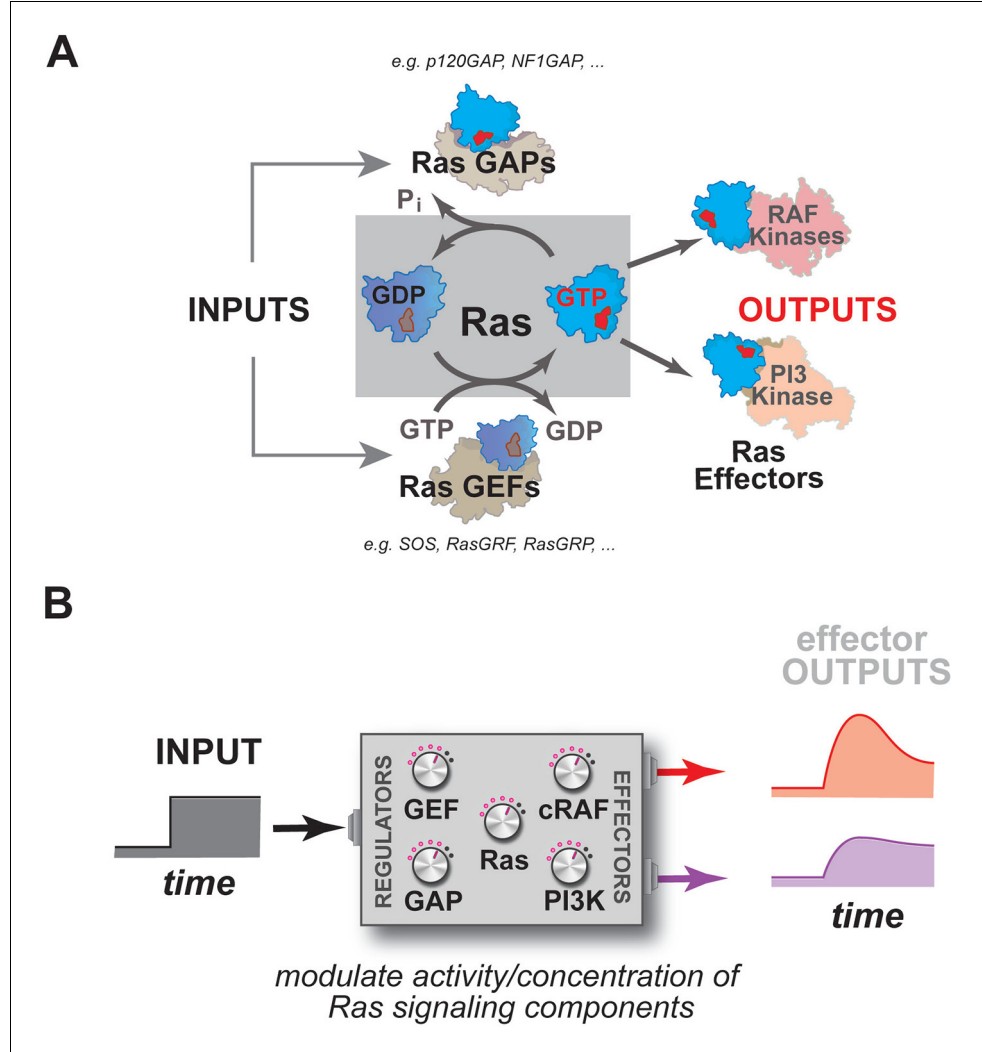

**Figure 1.** Multiple activities that are frequently perturbed in disease dynamically regulate Ras activity to control the assembly of downstream effectors during signal processing. (**A**) Depiction of the proximal architecture of Ras signaling systems. Ras is activated by guanine exchange factors (GEFs) that exchange GDP for GTP and is inactivated by GTPase-activating proteins (GAPs) that accelerate the hydrolysis of GTP. Activated Ras interacts with downstream effectors such as Raf or PI3 Kinase to assemble signaling complexes and elicit signaling outputs. (**B**) Abstraction of the proximal biochemical machinery underlying Ras processing of inputs into outputs, raising the question as to how the network configuration shapes signaling to multiple effector outputs.

By comparison, cell-based investigations of signaling are inherently multi-component and multi-turnover with respect to Ras, but provide far less control over the internal system parameters. In vivo assays usually contain fewer observables that are often far removed from the proximal signaling events. A variety of ingenious experiments have aimed to directly probe Ras activation during signaling in cells using fluorescent reporter molecules or super-resolution microscopy (*Rubio et al., 2010*; *Murakoshi et al., 2004*; *Nan, 2015*). However, the complications of using these reporter tools in living cells have made it difficult to systematically probe the behavior of the Ras signaling module. For example, it is difficult to know whether the response of one FRET reporter represents that of the diverse Ras effector species in the cell.

To date, there has been little *systems reconstitution* of Ras signaling – methodically exploring in vitro how the multiple activities that regulate Ras work together to dynamically cycle Ras and control the assembly of competing effectors on activated Ras during signal processing. As such, we know little about how the concentration and identity of the components within a Ras system define its

signaling properties. Understanding how these systems-level parameters shape behavior is critical, given that different cell types can harbor different configurations of network components (both in identity and expression levels) and also because distinct receptors may differentially activate key network components. In addition, the fact that many diseases are associated with perturbations to Ras and its associated regulators suggests that a systems level reconstitution of Ras signaling systems could be highly informative as to types of outputs and behaviors that these systems are capable of, as well as how these systems respond to perturbations such as mutation.

Here, to address this problem of systems reconstitution, we develop a new microscopy-based bead reconstitution assay of dynamic signal processing by human H-Ras, a canonical member of the Ras subfamily of GTPases. Our system includes both upstream regulators of Ras activity as well as multiple downstream effectors that bind to and perceive Ras•GTP signals. This allows us to follow multiple cycles of Ras turnover in real-time using the bead recruitment of fluorescently tagged effectors on Ras as the measured output – precisely analogous to the way cells measure and couple Ras•GTP levels to signaling outputs. In addition to giving an output that reflects a critical biological function (effector recruitment), this system allows us to precisely control the components and their concentration.

Using this system, we have explored how Ras signaling changes in response to network changes. We explore how oncogenic substitutions in Ras impact output behavior. We have scanned how the concentration of each type of network component sculpts effector outputs in response to a simple step input of GEF activity, how systems that contain multiple competing effector molecules behave, and how different mechanisms for implementing positive feedback reshape the landscape of output behaviors. More generally, the methods we develop herein provide a framework for studying the dynamics of other assembly driven signaling systems or more complex systems that incorporate multiple interconnected signaling nodes. This kind of analysis provides an overall design manual for Ras circuits, including those that could have originated through evolution or disease perturbation.

## Results

### Systems-level reconstitution of Ras signal processing in vitro: tracking effector output dynamics across multiple GTPase turnovers

To gain insight into the dynamics of how Ras transmits signals to downstream effectors under different network configurations or perturbations, we sought a dynamic in vitro reconstitution of Ras signal processing that would allow us to track effector outputs across multiple Ras turnovers. We reasoned that a microsphere surface charged with Ras could serve as a platform for the assembly and disassembly of fluorescent effector molecules from solution in response to inputs, much like the native Ras system (bound to the plasma membrane) functions in cells (*Figure 2A*). Signaling networks of defined composition could then be prepared from recombinant proteins and robust measurements of the dynamic output behavior could be determined by tracking the amount of effector on the surface over time for many individual beads and averaging their responses. Although such a system would not fully capture all biophysical features of cellular Ras signaling, such as GTPase diffusion in a fluid plasma membrane, partitioning between membrane microdomains, or GTPase exchange from the membrane, it serves as an excellent starting point to understand how these systems behave with fixed Ras molecules in a highly controlled setting (*Tian et al., 2007*; *Silvius et al., 2006*). Moreover, although the signaling activity of most Ras effectors is more complex than binding alone (see, for example: *Jelinek et al., 1996*; *Stokoe et al., 1994*), the regulated interaction of effectors with GTPase is the foundation on which any other complex signaling mechanisms will unfold, and thus represents a universal and fundamental feature of all Ras signaling systems that demands our understanding.

We first asked whether we could observe GTP-dependent translocation of an effector molecule to a Ras-coated bead catalyzed by a guanine nucleotide exchange factor (GEF). For our initial studies, we chose to use the catalytic domain from the RasGRF GEF, which is constitutively active and, unlike other GEFs, contains no allosteric feedback sites (*Freedman, 2006*). Ni-NTA microspheres were charged with a 16x-histidine tagged H-Ras•GDP (OFF state) that could not dissociate from the bead and incubated in the presence of 50 nM ($\sim K_D$) of a model effector: a fluorescently tagged Ras-binding domain (RBD) from the C-Raf kinase (*Block et al., 1996*). Under these basal conditions, the

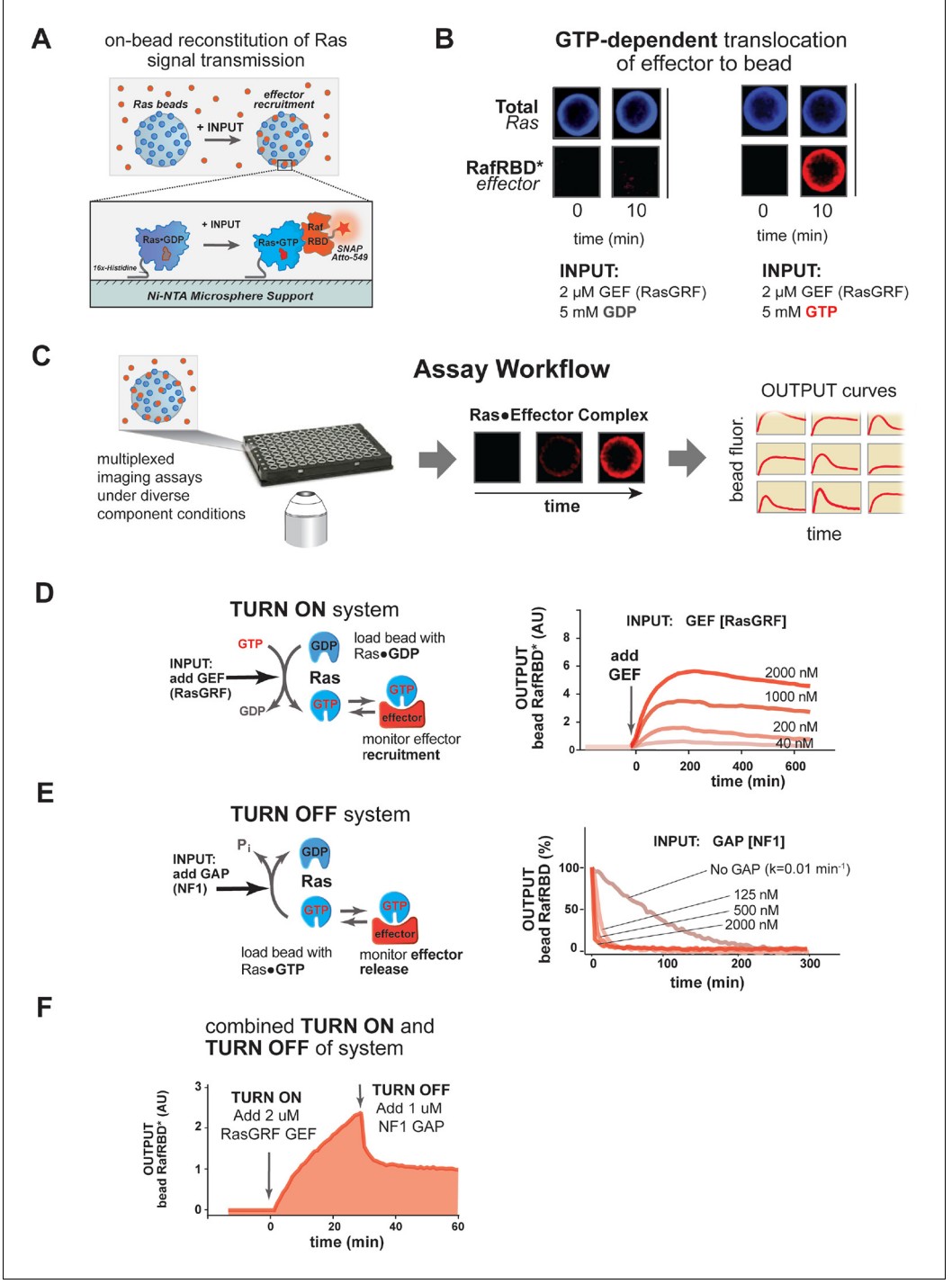

**Figure 2.** A network-level multi-turnover reconstitution of dynamic signal transmission from Ras to downstream effectors. (**A**) Bead-based approach used to study how Ras systems assemble effector complexes in response to inputs. By incubating Ni-NTA microspheres that have been loaded with Ras in solutions containing GEFs, GAPs, and fluorescent effectors, system outputs can be observed by monitoring the accumulation of effector on the bead-bound Ras. (**B**) Example of GEF-catalyzed GTP-dependent translocation of fluorescent effector to Ras-loaded bead. The amount of fluorescent effector bound to an individual bead before or after (10 min) addition of 2 μM GEF +/- 5 mM GDP or GTP is shown. (**C**) Schematic depicting multiplexed assay workflow in which the output dynamics for many different system configurations can be measured by microscopy. (**D**) Dose-dependent signaling response of effector translocation in response to increasing amounts of indicating RasGRF GEF activity. (**E**) Dose-dependent turn-off of output in the presence of saturating effector and increasing amounts of indicated

*Figure 2 continued on next page*

*Figure 2 continued*

NF1 GAP activity. (**F**) Combined turn on and turn off behavior of effector response when the system was activated with 2 µM RasGRF GEF and after 30 min NF1-GAP was added. GAPs, GTPase-activating proteins; GEFs, guanine exchange factors.

The following figure supplement is available for figure 2:

**Figure supplement 1.** RasGRF GEF and NF1 GAP dose-dependent effects on effector output behaviors.

amount of fluorescence on the bead was comparable to the background levels of fluorescence from the effector in solution. We then added as input 2 µM of the catalytic domain of the RasGRF GEF and 5 mM of either GDP or GTP and monitored the output of effector fluorescence on the bead (*Figure 2B*). This amount of nucleotide in solution is in vast excess of the small amount of bead-bound Ras present in the reactions, providing essentially an infinite supply of nucleotide for these reactions on the timescale we examine (detailed in 'Materials and methods'). Upon GEF and nucleotide addition, there was noticeable accumulation of fluorescent effector on the bead surface of the GTP containing reactions within seconds, and considerable fluorescent signal was observed by 10 min. In contrast, no fluorescent effector accumulated on the surface of reactions containing GDP, indicating that GEF-catalyzed translocation of the effector was dependent on Ras becoming GTP loaded. Having seen GTP-dependent GEF-catalyzed translocation of an effector to a Ras-charged bead surface, we were now in position to prepare signaling networks of arbitrary configuration and assay their output dynamics in multiplex using our microscopy-based assay (*Figure 2C*).

We first asked whether we could observe *quantitative* differences in the system's output behavior when different amounts of GEF activity were used as inputs. When identically loaded beads were stimulated with increasing amounts of RasGRF, we observed both faster rates of effector translocation and higher steady state amplitudes of effector output (*Figure 2D*, *Figure 2—figure supplement 1A–B*). Initial rates of effector translocation appeared to show a hyperbolic response to increasing GEF, while steady states effector levels showed a linear response. Together, these data demonstrate that our reconstituted Ras signal processing system produced outputs that responded quantitatively to the amount of GEF present in the system.

In addition to being able to turn on, a *dynamic* reconstitution of Ras signal processing must be reversible and be able to turn off. To test the reversibility and turn-off of our system, we prepared beads loaded with H-Ras•GTP (ON state), incubated them with a saturating excess amount of C-Raf RBD effector (2.5 µM total, 50 nM fluorescently labeled), and monitored the loss of effector signal from the bead over time (*Figure 2E*). In the absence of any GAP, effector fluorescence decayed with a rate constant of 0.01 min$^{-1}$, which is similar to the expected rate of intrinsic hydrolysis by H-Ras under our assay conditions (*Neal et al., 1988*; *Gibbs et al., 1984*). This result is consistent with previous observations that intrinsic hydrolysis by the GTPase continues to occur even in the presence of saturating amounts of effector (*Herrmann et al., 1995*). When increasing amounts of the catalytic domain from purified Neurofibromin-1 GAP (NF1-GAP [*Scheffzek et al., 1998*; *Xu et al., 1990*]) were included in the reactions, effector signal disappeared from the bead at an increased rate in a dose-dependent manner with a hyperbolic dependence on the GAP concentration (*Figure 2E*, *Figure 2—figure supplement 1C*). Thus, as with the turn-on of the system, these data indicate that our reconstituted Ras signal processing system displays turn-off that responds quantitatively to the amount of GAP present in the system.

Having found that our system can produce effector outputs that are turned on by GEF and turned off by GAP, we wanted to verify that the system was truly multi-turnover and that the output dynamics would respond to both these activities working in concert. To this end, we incubated Ras•GDP-loaded beads with 50 nM fluorescent effector, initiated signaling with 2 µM RasGRF GEF and 5 mM GTP, and then added 1 µM NF1-GAP at 25 min (*Figure 2F*). As before, addition of GEF stimulated assembly of effector on the bead as the levels of Ras•GTP increased. When NF1-GAP was added to the reactions, the system responded with a rapid decrease in effector levels before stabilizing at a non-zero plateau corresponding to the non-equilibrium steady state maintained by the balance of effector, GEF and GAP activities present in the reaction.

Taken together, these data imply that our on bead reconstitution of H-Ras signal processing can semi-quantitatively track dynamic effector outputs across multiple cycles of Ras activation and deactivation during signaling. This system now puts us in position to explore how different mutational states, network configurations, protein identities, or feedback mechanisms affect signal processing by Ras GTPase systems.

## Distortion of signaling by oncogenic Ras alleles depends on balance of positive and negative regulatory activities in the network

Mutations of Ras (especially at the G12, G13, or Q61 positions) are frequently associated with cancer or other diseases (*Barbacid, 1987*). These alleles are primarily thought to impact Ras signaling through three mechanisms: 1) decreasing the intrinsic hydrolysis rate of the GTPase, 2) blocking GAP-mediated hydrolysis of the GTPase, and 3) potentially altering the interaction and preference of the GTPase for downstream effectors (*Figure 3A*) (*Trahey and McCormick, 1987*; *Barbacid, 1987*; *Rajalingam et al., 2007*; *Smith and Ikura, 2014*). The same mutant allele of Ras can elicit different phenotypes in different cell types and tissues. Thus, we wanted to use our in vitro systems reconstitution assay to determine which system configurations are most sensitive to these oncogenic perturbations.

Using our dynamic, multi-turnover reconstitution of Ras signal processing, we examined how signaling networks bearing the G12V allele of the Ras GTPase distorted effector outputs relative to the wild-type Ras GTPase. By labeling wild type and G12V Ras GTPases with different fluorophores, we could distinguish beads loaded with each variant in a common solution of network components to see differences in effector outputs from each system side-by-side.

For these and future experiments, we display the output response data in two ways: (1) we show the absolute response, which conveys information about both the *amplitude and the shape* of the output response, and (2) we show the responses after normalizing to the maximum value of the response, which conveys information *only about the shape* or dynamic profile of the output. The latter is particularly useful for seeing how the shape of two responses differs when the amplitudes are substantially different.

With this approach, we first examined the output of 50 nM C-Raf RBD effector from G12V or wild-type Ras networks *without* GAP activity in response to a step input of 2 µM of the GEF RasGRF (*Figure 3B*, *Video 1*). Under this network configuration, wild type and G12V Ras systems produced very similar outputs with almost no difference in the total integrated effector output and only small differences in the overall dynamics of their responses. This suggests that neither the intrinsic hydrolysis nor changes in C-Raf RBD effector interactions of the G12V substitution is particularly perturbative to the output of the signaling system under this GAP-free network configuration.

We then looked at the system responses of wild type and G12V Ras systems to the exact same step-input (2 µM RasGRF GEF) but in networks that now included 1 µM basal NF1-GAP (*Figure 3C*, *Video 2*). Unlike in the GEF-only networks, both the dynamics and amplitude of the effector output were substantially distorted by the Ras-G12V allele. In this network configuration, wild-type Ras produced a *transient* response that peaked within an hour and declined to a steady state less than 20% its maximum value. In contrast, outputs from G12V were sustained and increased in magnitude for over 6 hr before settling at a steady state more than 40 times higher than wild-type Ras. Thus, the G12V mutation is significantly perturbative in a high-GAP network context.

Together these data and model imply that the balance of positive and negative regulatory activities in a signaling network impacts the severity by which Ras-G12V distorts signals. Similar results were also observed for G12C and Q61L alleles (*Figure 3—figure supplement 1*). To determine which particular configurations are most distorted by the G12V allele, we measured the effector output response across four different input strengths ([GEF] activity) and four different NF1-GAP levels. We then calculated a distortion score as the fold-change integrated output from Ras-G12V relative to wild-type Ras and interpolated the responses from these 16 configurations to produce a phase diagram of signal distortion by G12V under different network conditions (*Figure 3D*). This revealed that G12V alleles were most perturbative with low-GEF inputs and a high-GAP network context, conditions in which the GAP activity would, for wild-type Ras, completely dominate over the small amount of activating GEF input. These observations are consistent with models of oncogenic Ras signaling in which low-level inputs or noise from the environment that would normally be filtered out by basal GAP-activity are misinterpreted by the cell as *bona fide* activating signals.

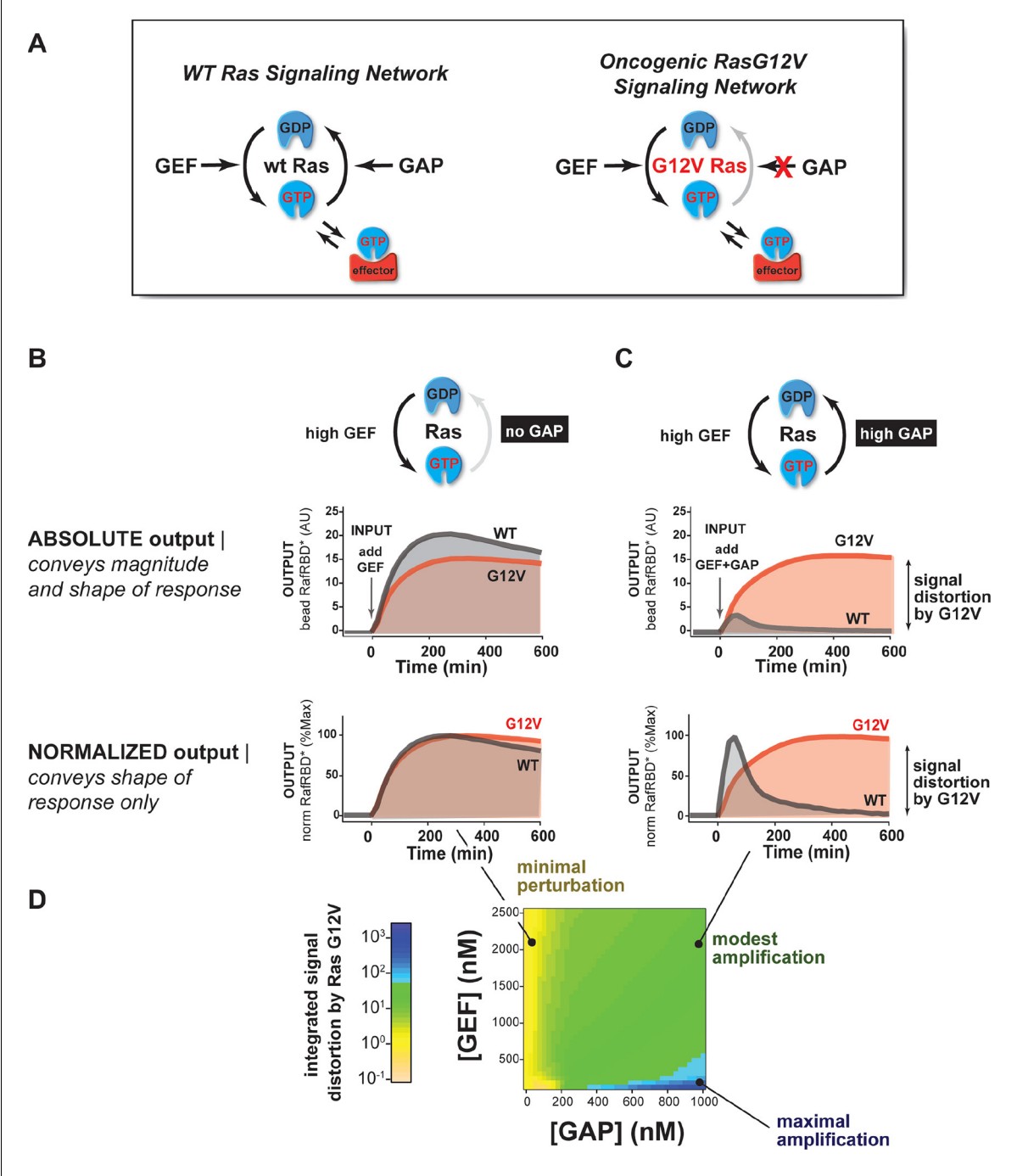

**Figure 3.** The extent of signal processing distortion by oncogenic alleles of Ras depends on the balance of positive and negative regulatory activities in the network. (A) Depiction of wild-type (WT) Ras and oncogenic G12V Ras, illustrating the modes by which mutation is thought to impact the network behavior: changing intrinsic hydrolysis rate, blocking GAP-mediated hydrolysis, and modulating effector interactions. (B) Absolute and normalized effector responses to a 2 µM RasGRF GEF step input in the absence of any GAP activity. (C) Absolute and normalized responses of the same step input as in (B), but with 1 µM NF1 GAP activity present in the network. (D) Experimentally determined phase diagram derived from 16 output responses showing the magnitude of signal distortion caused by G12V substitution (defined as fold-change in integrated signal of G12V relative to WT) in different GEF and GAP network configurations. GAP, GTPase-activating protein; GEF, guanine exchange factor.
The following figure supplements are available for figure 3:

**Figure supplement 1.** The extent of signal processing distortion by oncogenic G12C and Q61L alleles of Ras depends on the balance of positive and negative regulatory activities in the network.

*Figure 3 continued*

**Figure supplement 2.** Kinetic modeling and simulations suggest competition and intermediate GTPase states contribute to transient system behavior.

## Modeling suggests competition and intermediates contribute to transient signaling dynamics of wild-type Ras-GTPase systems in contrast to oncogenic variants

The transient response of the wild-type GTPase in the high-GAP network context was unanticipated, as this phenomenon cannot be explained by the simplest model of effector/GTPase binding in which the GTPase toggles ON and OFF with rates directly proportional to [GAP] and [GEF] (*Figure 3—figure supplement 2*). Indeed, this is consistent with analytic results that state that two-state systems cannot show overshoot behavior (*Jia et al., 2014*). However, transient overshoot could be easily introduced into the system by two non-mutually exclusive mechanisms: (1) competition between effectors and GAP molecules; and (2) the existence of a post-hydrolysis GTPase state that is refractory to GEF stimulation.

We found that extending the two-state model to include competition could produce overshoot, but this required higher concentrations of the Raf-RBD effector than the GAP and tighter binding of GAP to GTPase than Raf-RBD to GTPase, neither of which are consistent with the known parameters of the system (*Figure 3—figure supplement 2*). Moreover, because GAP can bind (but not stimulate the hydrolysis of) the G12V, competition cannot explain the differences that we observed between the wild-type and G12V dynamics.

In contrast, the introduction of a refractory GTPase state to the model produced overshoot that captured the key features of the observed data within physiological parameters: a monotonic response to inputs in low-GAP networks but a transient response in high-GAP networks (*Figure 3—figure supplement 2*). It also is consistent with the differences in dynamics that we observed between G12V and wild-type Ras. While the exact molecular nature of this intermediate GTPase state is not clear at present, this does not hinder our ability to study its consequences for signal processing.

## Concentration and identity of Ras network components modulates timing, duration, shape, and amplitude of effector outputs

Our comparison of wild type and G12V Ras signaling systems and our associated model illustrated the importance of the network composition in shaping signal processing outputs. Each individual network component is, in essence, a separate 'dial' of the Ras signaling system that can be turned by adjusting the concentration of that component (*Figure 4A*). Because expression levels of signaling components vary across different cell types and are often different in oncogenic states, we

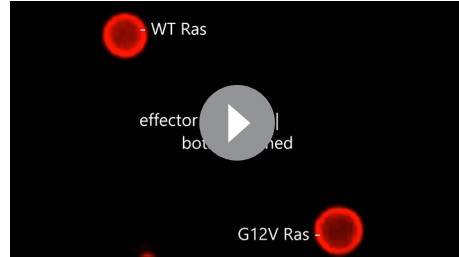

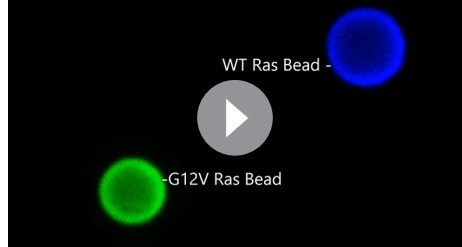

**Video 1.** Response of wild type and G12V Ras systems in GAP-free network context. The effector output (red) from a representative bead loaded with wild-type Ras (blue) or G12V Ras (green) is shown. 2 μM RasGRF was provided as an activating input. Time-steps are separated by 15 min. Associated with data in main-text *Figure 3B*.

**Video 2.** Response of wild type and G12V Ras systems in high-GAP network context. The effector output (red) from a representative bead loaded with wild-type Ras (blue) or G12V Ras (green) is shown. 2 μM RasGRF was provided as an activating input and the system contained 1 μM NF1-GAP. Time-steps are separated by 15 min. Associated with data in main-text *Figure 3C*.

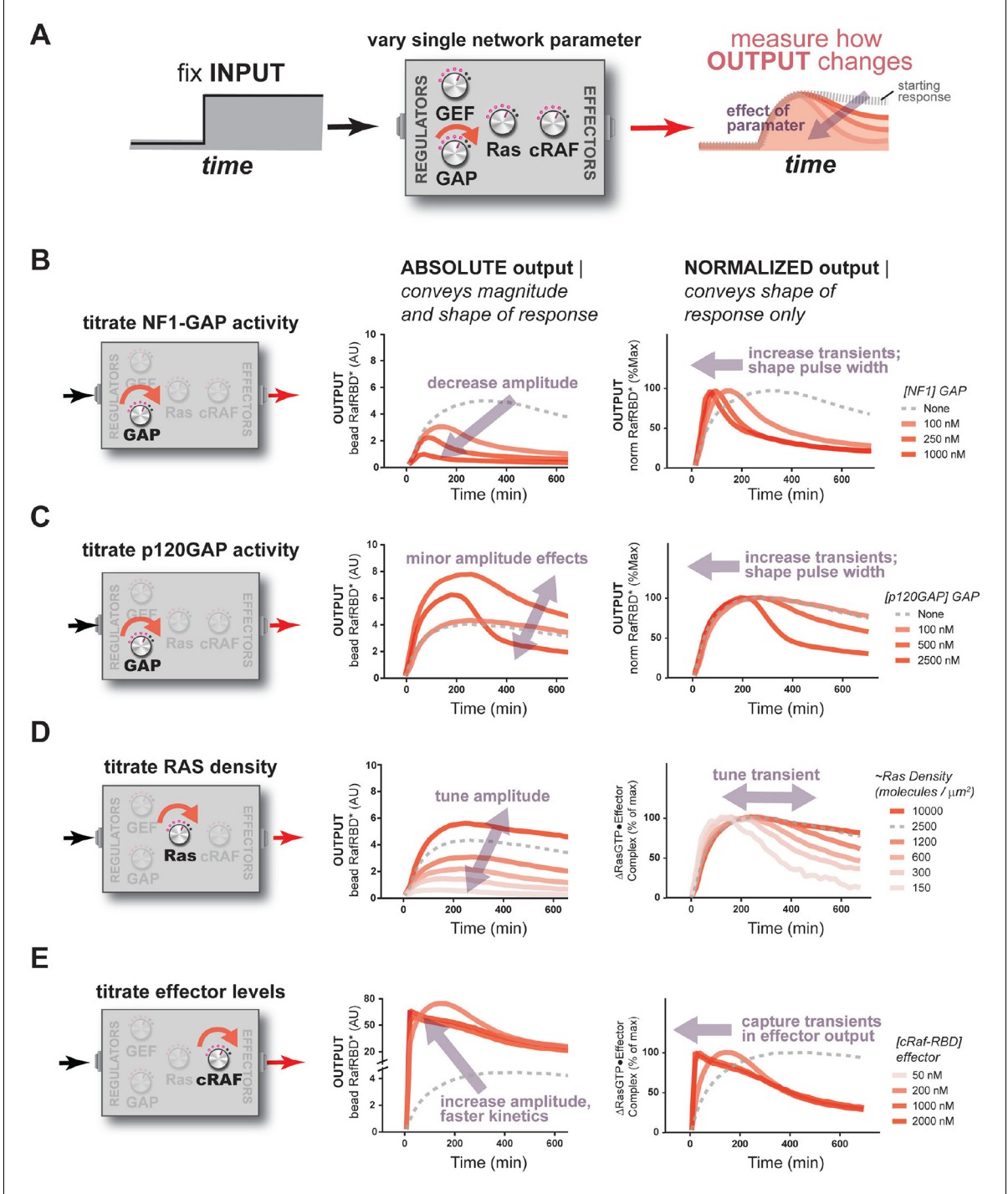

**Figure 4.** The concentration and identity of each Ras network component can modulate the timing, duration, shape, or amplitude of effector outputs. (**A**) Depiction of the experimental setup: a fixed step-input is applied to a panel of Ras signaling systems in which the concentration of a single network component is varied to determine how each network component individually modulates system output. (**B**) Absolute and normalized effector responses to step-input in the presence of increasing amounts of the NF1 gap. (**C**) Absolute and normalized effector responses to step-input in the presence of increasing amounts of the p120 GAP. (**D**) Absolute and normalized responses to step-input in the presence of different densities of Ras on the bead surface. (**E**) Absolute and normalized responses to step-input in the presence of increasing amounts of the C-Raf RBD effector. GAP, GTPase-activating protein; RBD, Ras-binding domain.

wondered how the level of each network component impacted signal processing wild-type Ras signaling networks. To this end, we fixed a particular input (2 µM RasGRF GEF) and, starting from a particular initial system configuration (~2500 Ras molecules × µm², 50 nM C-Raf effector, no GAP activity) with an associated output response, asked how titration of individual system components modulated the effector output.

## Effects of GAP activity: the properties of distinct GAP species

Having seen dramatic effects of negative regulatory activities in our distortion analysis of G12V Ras, we first looked more generally at how GAP activity sculpted signal processing dynamics in wild-type Ras networks. We considered two distinct GAPs domains with different biochemical properties and expression patterns: the NF1-GAP and the p120-GAP. NF1-GAP, which is expressed somewhat ubiquitously but highest in neuronal cells and leukocytes, has a tight $K_M$ for Ras (0.3 µM) and modest $k_{cat}$ (1.4 s$^{-1}$) (*Wiesmüller and Wittinghofer, 1992*). In contrast, p120GAP has a higher $K_M$ (9.7 µM) for Ras, but also a higher $k_{cat}$ (19 s$^{-1}$), and shows a much more ubiquitous expression profile (*Wiesmüller and Wittinghofer, 1992*).

When increasing amounts of NF1-GAP were included in networks, the same step input of GEF activity produced dramatically different effector output behaviors (*Figure 4B*). First, increasing levels of NF1-GAP resulted in substantially lower end point steady state levels of bound effector. Moreover, the amount of NF1-GAP changed the *dynamics of approach* to steady state significantly: while GAP-free networks approached steady state from below, networks containing NF1-GAP showed transient overshoot that decayed back down to the steady state from above, creating an initial pulse of strong output followed by weaker levels of output in the long term. The pulse-width and peak-time of effector outputs were inversely correlated with the concentration of the NF1-GAP (*Figure 4B*). Thus, NF1-GAP shapes not only the final steady-state levels of output in the system, but also the shape, timing and duration of Ras signal processing.

As discussed previously in the context of our initial kinetic model, overshoot dynamics such as these are a hallmark of energetically-driven chemical systems that contain *more* than two states approaching a non-equilibrium steady-state, in which intermediate system states can transiently accumulate and then de-accumulate as the system begins to cycle (*Jia et al., 2014*; *Hong, 2014*; *von Bertalanffy, 1950*). This is analogous to the way in which enzyme intermediates can transiently accumulate in the pre-steady state depending on the rate-constants for the microscopic steps, for example if product release is rate-limiting (*Fersht, 1999*). Our observation of these phenomena in Ras circuits is only possible because our system uses bona fide GTP that allows for Ras cycling to actively occur. Consistent with this, the G12V, G12C, and Q61L mutants we analyzed previously – which cannot efficiently cycle – did not show such overshoot behavior even under the highest GAP conditions examined.

We repeated this analysis using the catalytic domain of p120GAP in our networks instead of NF1-GAP. Compared to the NF1-GAP, the impact of the p120GAP on the end-point effector output of the system was much less substantial (*Figure 4C*). This may in part be owing to the much higher $K_M$ of p120GAP compared to NF1-GAP, leading to much lower effective GAP activity in the concentration regimes we could readily explore. Nonetheless, increasing amounts of p120GAP levels did lead to a marked change in the output dynamics of Ras signal processing in a manner similar to that of NF1-GAP and is consistent with the consequences of altering the $K_M$ and $k_{cat}$ of the GAP in our kinetic model (*Figure 5—figure supplement 3*). As with NF1-GAP, transient behaviors emerged when p120GAP was present, and the pulse-width and peak-time of effector outputs were inversely correlated with the concentration of p120GAP. Thus, like NF1-GAP, p120GAP shapes the dynamics and steady-state behavior of signal processing by Ras, but with a different dose-dependent behavior owing to its unique biochemical characteristics.

## Effects of Ras density: how expression level and clustering can alter signaling

Given that Ras expression level can vary among different cell types and that Ras distribution in the plasma membrane can be both free as well as packed into high-density nanoclusters (*Janosi et al., 2012*; *Plowman et al., 2005*), the next system parameter we considered was the density of Ras. We made a dilution series of Ras, loaded beads with each dilution, and then mixed these beads together

to assay the responses of different Ras densities side-by-side in the exact same network solution. Because the Ras was fluorescently labeled, we could estimate the Ras density from the intensity of each bead and bin the responses from similar density beads together to obtain average behaviors for different density classes.

Applying this approach to our fixed step-input, we found that the Ras density was directly correlated with the amplitude of effector output produced by the system (*Figure 4D*). This is expected as higher Ras densities provide more molecules that can form Ras•Effector complexes. Additionally, the normalized traces of these responses revealed differences in the dynamic behavior of effector output. When Ras densities were high, the step-response effector output monotonically approached its steady state and was sustained.

At lower Ras densities, however, effector output responses were increasingly transient in character. These outputs peaked early in the response and then decreased significantly to a lower value over the time-course. These differences likely reflect a switch from a network configuration in which Ras is in excess of the GEF to one in which the GEF is in excess of Ras, and this idea is supported by our kinetic model (*Figure 5—figure supplement 3*). Intuitively, these differences will cause a change the initial fraction of the Ras population that is activated, such that a much larger synchronous cohort is formed at lower Ras densities.

## Effects of C-Raf RBD effector concentration: active roles for downstream components

Finally, we considered the impact of the concentration of effectors, the molecules that are used by cells to perceive and interpret Ras•GTP dynamics in the cell, on signal processing. For these experiments, we fixed the amount of fluorescent C-Raf RBD effector at 50 nM and added additional unlabeled C-Raf RBD to achieve a target final concentration of effector. We could then normalize the observed fluorescent effector output by its proportion in the total effector population to infer the true magnitude of the output.

When we measured the system step-response in the presence of increasing amounts of effector in the network, dramatic changes in the both the amplitude and dynamics of output were observed (*Figure 4E*). At an effector concentration of 50 nM ($\sim K_D$), the system output showed the typical monotonic approach to a sustained steady-state.

When we increased effector concentrations, both the amplitude and the dynamics of the output response changed markedly. At 250 nM effector concentration, output increased to a level *20 times* that of the 50 nM effector system (measured at 120 min) at that time, before decaying down to its final steady state level. At even higher effector concentrations (500 nM, 1000 nM), system output peaked quickly within 20 min and then decayed monotonically over several hours to the final steady state level. These data demonstrate that higher effector concentrations not only increase output amplitudes but also enable the output to capture more transient features of the upstream Ras•GTP signal. This idea is supported by our kinetic model (*Figure 5—figure supplement 3*) and makes sense intuitively because higher effector concentrations decrease the time needed to equilibrate against a fixed concentration of Ras•GTP; if this process is too slow, transient aspects of the GTPase activation/deactivation dynamics that occur on a faster timescale will be missed in the effector output. This implies that effectors are not merely passive conduits for transmitting upstream Ras dynamics, but instead play an active role in interpreting and perceiving what features of those dynamics to pass downstream during signaling. A corollary of this observation is that erroneous overexpression of effector molecules not only can lead to higher amplitude outputs, but also can drastically alter the overall dynamics of the system behavior as well.

## Diverse dynamic outputs achieved by titration of components

Our data show that the identity and concentration of each component in a Ras signaling network can have a profound impact on the timing, duration, shape, or amplitude of effector outputs. This implies by tuning the abundance and identity of network components and controlling the strength of inputs, a variety of diverse dynamic effector output programs can be realized by Ras signaling system (*Figure 5A*). To explore this space of output programs more thoroughly, we fixed a particular effector concentration (50 nM, $\sim K_D$) and measured system output from different Ras densities, p120GAP concentrations, and input strengths ([GEF]). In total, we experimentally measured output

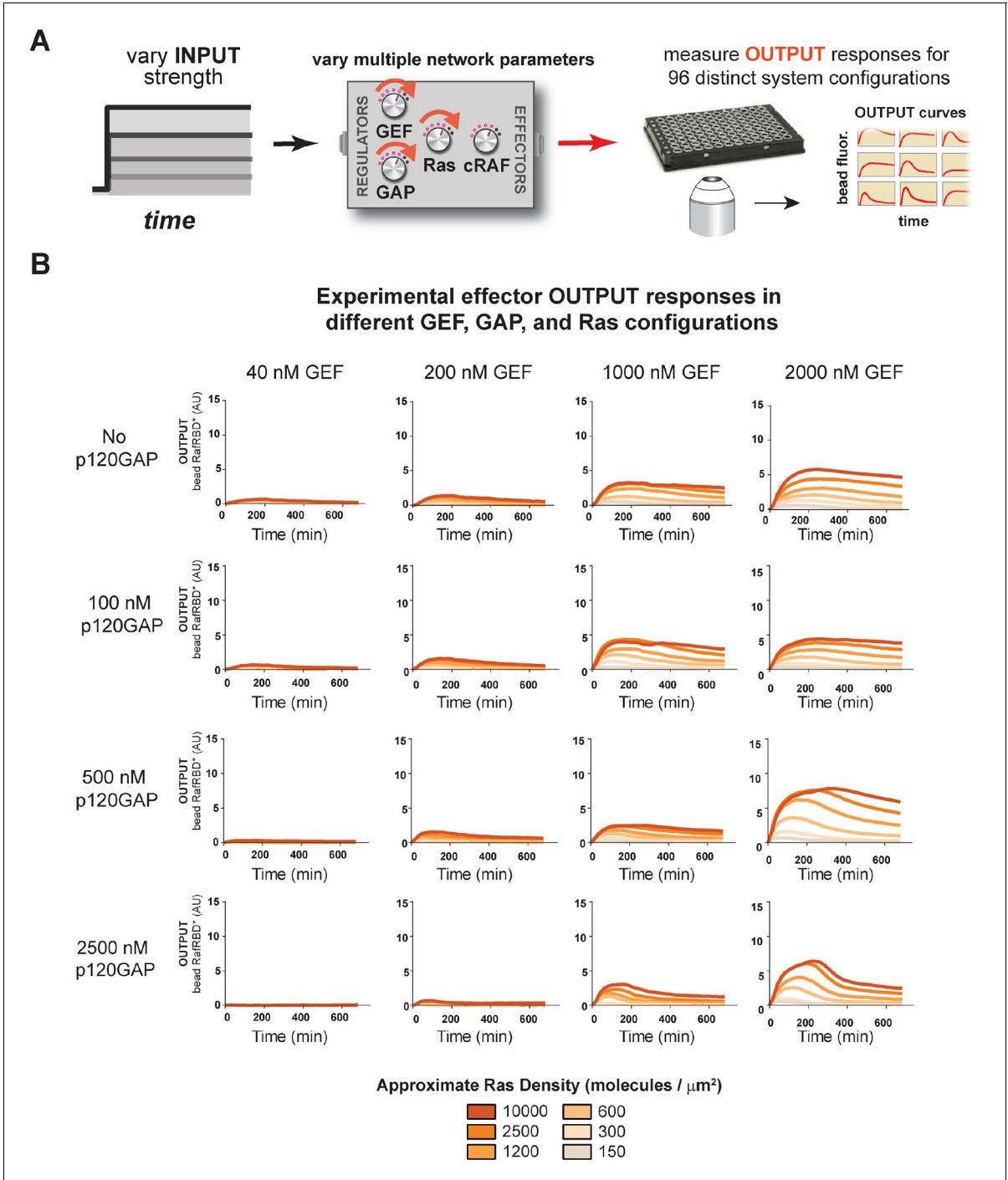

**Figure 5.** Tuning the levels of GEF, GAP, and GTPase provide access to a rich and diverse space of possible Ras signal processing behaviors. (**A**) Depiction of the experimental setup: four different inputs (changes in apparent GEF activity) are applied to a panel of Ras signaling systems sampling four different p120GAP concentrations, and six different Ras densities resulting in experimentally determined output responses for 96 different system configurations. (**B**) Experimentally determined absolute effector OUTPUT responses across 96 different system configurations. Each graph corresponds to a particular GEF/GAP configuration, and each of the curves within that plot corresponds to a different Ras density as indicated by the color of the curve. GAP, GTPase-activating protein; GEF, guanine exchange factor.

The following figure supplements are available for figure 5:

**Figure supplement 1.** Normalized (to maximum output) responses of p120GAP/RasGRF/RafRBD/Ras signaling system under a variety of network configurations.

*Figure 5 continued on next page*

*Figure 5 continued*

**Figure supplement 2.** Structure of RasGRF/p120GAP/H-Ras/cRaf response space determined from outputs of 96 system configurations.

**Figure supplement 3.** Kinetic modeling and simulations are consistent with experimental observations about how system behavior is influenced by network configuration.

responses for 96 configurations corresponding to four different input strengths, four different GAP concentrations, and six different Ras densities (*Figure 5B*, normalized responses in *Figure 5—figure supplement 1*).

The diversity of dynamic output responses we observe highlights the versatility and tunability of the Ras signaling system: sustained responses of arbitrary amplitudes can be produced as well as transient responses with different peak times and magnitudes of overshoot, all by simply by co-varying the levels of different system parameters. Because the number of output responses we measured is large, we extracted three features from each output response trace that describe the behavior – integrated signal intensity, initial rate of activation, and a transient score that reflect the amount of overshoot in the response – and interpolated these values for three different Ras density bins to create an experimentally determined signal processing phase diagram for each output feature as a function of the network configuration (*Figure 5—figure supplement 2*).

These phase-diagrams not only summarize the output responses we measure but also clarify the structure of the space of Ras signaling behaviors. This provides a roadmap for understanding how output responses change as we alter system parameters and helps predict the impact of perturbations that move the system from one region of the space to another. The isoclines in these diagrams also highlight the existence of different network configurations with equivalent signaling behaviors. These correspond to neutral paths in network-space that the signaling system can drift along without immediate consequence to signaling output.

## Different Ras effectors perceive the same input uniquely, enabling multiple distinct temporal outputs to be encoded in multi-effector networks

So far, our characterization of Ras signal processing has used the C-Raf RBD as the sole downstream effector, but in living cells these networks typically contain *multiple* effectors targeting *different* output responses that are in *competition* with one another for access to activated Ras, with each of these effectors possessing its own affinity for Ras•GTP and expression level in the cell (*Herrmann, 2003*; *Smith and Ikura, 2014*). Indeed, our kinetic modeling implied that competition was another important source of dynamic complexity in these systems (*Figure 3—figure supplement 2*). Because our reconstituted signal processing is microscopy based, we can track the behavior of multiple distinct competing effectors processing signals on the same bead simultaneously by labeling each effector with a different color fluorophore (*Figure 6A*). To this end, we purified and labeled RBDs from the A-Raf and B-Raf kinases, which have lower ($k_{off}$ = 5.52x10$^{-4}$ s$^{-1}$, $k_{on}$ = 7.20x10$^{3}$ M$^{-1}$s$^{-1}$) and higher ($k_{off}$ = 1.48x10$^{-4}$ s$^{-1}$, $k_{on}$ = 1.32x10$^{4}$ M$^{-1}$s$^{-1}$) affinities for Ras•GTP than C-Raf ($k_{off}$ = 2.15x10$^{-4}$ s$^{-1}$, $k_{on}$ = 1.02x10$^{4}$ M$^{-1}$s$^{-1}$), respectively (*Fischer et al., 2007*), to examine the signal processing behavior of two-effector systems in either GAP-free or high NF1-GAP networks.

We first considered networks containing equivalent, physiological amounts of C-Raf and B-Raf effectors, which both have high affinity for Ras•GTP (*Smith and Ikura, 2014*). In response to a 2 μM RasGRF GEF step input in a GAP-free network, C-Raf and B-Raf processed these signals with different amplitudes and completely different dynamics (*Figure 6B*). Initially, C-Raf and B-Raf outputs assembled at comparable rates, but within 1 hr C-Raf output peaked and began to decrease while B-Raf continued to increase in output monotonically over the entire time course. When these step-responses were re-examined in a high NF1-GAP network context, we continued to see different effector responses between C-Raf and B-Raf: C-Raf output peaked within 30 min before sharply declining to a steady state value 25% of its maximum. In contrast, B-Raf output peaked later at 1 hr, and declined to a 75% its peak maximum, a much higher steady state compared to C-Raf

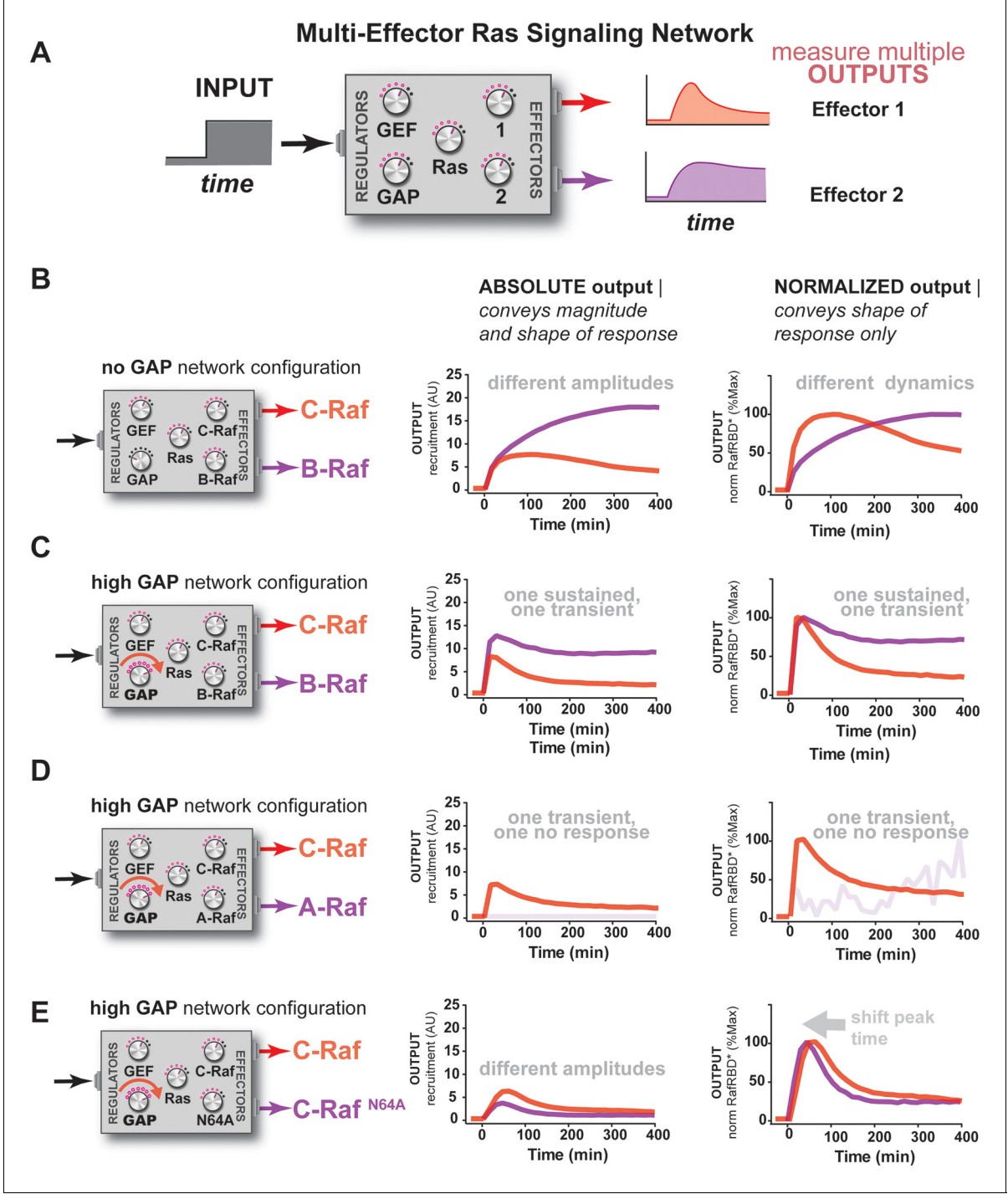

**Figure 6.** Unique interpretation of Ras•GTP signals by different effectors in multi-effector networks encodes multiple distinct temporal outputs in the system response. (**A**) Depiction of the experimental design: a fixed step-input is applied to a particular network configurations in which more than one effector molecule is, resulting in multiple simultaneous system outputs that are measured. (**B**) Absolute and normalized responses to step-input of C-Raf RBD and B-Raf RBD in the absence of any GAP activity. (**C**) as in (**B**) but with 1 µM NF1-GAP present in the signaling network. (**D**) Absolute and normalized responses to step-input of C-Raf RBD and A-Raf RBD with 1 µM NF1-GAP present in the signaling network. (**E**) Absolute and normalized responses to step-input of C-Raf RBD and the C-Raf^N64A mutant RBD with 1 µM NF1-GAP present in the signaling network. RBD, Ras-binding domain

The following figure supplements are available for figure 6:

**Figure supplement 1.** Additional examples of how the unique interpretation of Ras•GTP signals by different effectors in multi-effector networks encodes multiple distinct temporal outputs in the system response.

*Figure 6 continued on next page*

*Figure 6 continued*

**Figure supplement 2.** Kinetic modeling and simulations show that competition between effectors allows multiple temporal responses to be encoded in the system output.

(*Figure 6C*). Thus, in this case, one effector (B-Raf) produces a transient response while a different effector (C-Raf) produces a more sustained output.

Why do different effector molecules interpret upstream Ras signals with different output dynamics as we observed? Two factors contribute to this phenomenon. First, as we previously saw with one-effector systems, the time it takes for a binding process to equilibrate impacts the extent to which it can track the transient dynamics of its target. Distinct effector molecules have different binding affinities that are determined by different on and off rates from the target, and thus will interpret Ras dynamics differently. Second, these effector molecules are in competition with one another for access to the supply of time-varying activated Ras. Because we are observing the nonequilibrium binding dynamics of effectors to Ras, an effector may be competitive in the short-term (kinetically determined phase) but less so in the long-term (equilibrium-determined phase) (*Motulsky and Mahan, 1984*). Appropriately, the inclusion of competing effectors with physiological parameters in our model produced dynamics recapitulating such different effector responses (*Figure 6—figure supplement 2*).

We then performed a similar analysis for two-effector systems containing equivalent, physiological amounts of C-Raf and A-Raf RBDs, where A-Raf has a weaker affinity for Ras•GTP and higher off-rate from the GTPase than C-Raf (*Smith and Ikura, 2014*). In GAP-free networks, the step-responses for A-Raf and C-Raf differed most significantly in terms of their amplitudes, with A-Raf output 20 times lower than C-Raf output (*Figure 6—figure supplement 1*). The normalized traces of each effector output show only minor differences in the dynamics of these outputs. In high NF1-GAP networks, we could only detect an output response from the C-Raf effector; the ability of A-Raf to assemble on the bead was either filtered out by the low affinity combined with high levels of GAP activity, or was simply too weak to detect above the background A-Raf in solution (*Figure 6D*).

Clearly, effector molecules with distinct identities result in differential interpretation of inputs, but how easy is it for a given effector molecule to acquire a new dynamic behavior? Given that A-Raf, B-Raf, and C-Raf have different affinities for the GTPase, we wondered whether mutations in C-Raf that changed its affinity would be sufficient to drive new dynamic behaviors. To this end, we characterized the step-response of two-effector network containing equivalent amounts (50 nM) of the C-Raf RBD and the C-Raf$^{N64A}$ point mutant that has decreased affinity for Ras•GTP (*Block et al., 1996*). Different dynamic output behaviors for each effector were observed for both GAP-free (*Figure 6—figure supplement 1*) and high NF1-gap (*Figure 6G*) networks. In both contexts, the lower affinity C-Raf N64A mutant peaked earlier and had lower overall amplitude than the wild-type C-Raf RBD, and like before, high GAP-networks accentuated these temporal differences. Consistent with this, we found that parameter changes as small as 2-fold could produce subtle shifts in the timing of effector reponses of our kinetic model (*Figure 6—figure supplement 2*). This shows that new dynamic behaviors are readily realized by mutation of an effector molecule.

Taken together, this analysis shows that distinct effector molecules can perceive the same input to a Ras signaling system with different dynamics and amplitudes depending on their affinities and biochemical properties. Consequently, a single-step input can be in principle be used encode multiple classes of temporally distinct outputs that peak and decline out of phase with one another, allowing for a sequence of different activities to be organized during signal processing. For example, we were able to produce a three-wave activation response of three distinct effectors in our kinetic model by simply modifying concentrations and off-rates (*Figure 6—figure supplement 2*). Furthermore, the context of other regulators (e.g. extent of GAP activity in the network) can influence how these different dynamic responses unfold, magnifying temporal distinctions in some cases while restricting the ability of certain effectors to assemble productively at all in other cases. Finally, because even simple point mutations to an effector can dramatically alter its output dynamics, new dynamic patterns are not difficult to produce and can be easily accessed during evolution.

## Positive feedback (GTPase$_{ON}$→GEF) in Ras networks alters signal processing behavior in different ways depending on how the feedback mechanism is implemented

The signaling networks we have examined thus far are solely the product of constitutive enzymatic activities and effector assembly processes unfolding in the simplest possible Ras GTPase signaling circuit. Our analysis found that in high-GAP systems, the 'ground state' output for a step-response will transiently overshoot the final steady state. In some instances, this behavior could be useful for the cell, for example to create an adaptive response or to produce distinct temporal phases in multiple downstream effector outputs; in other instances this overshoot behavior could prove undesirable, for example if the overshoot provoked a proliferative response to non-proliferative level of input.

Many cellular circuits modulate intrinsic behaviors of a signaling system by including additional layers of regulation and feedback control that could alter the signaling properties of the system. To gain insight into how such regulation might alter the ground state signaling behavior of Ras GTPase systems, we examined the effect of introducing two different modes of GTPase→GEF positive feedback (defined as active Ras promoting more activation of Ras) on system signaling behavior.

### Recruitment-based feedback mechanism (GTPaseON → GEF localization) amplifies weak inputs

One common mode of generating positive feedback in signaling is through recruitment. For example, in yeast, the GTPase Cdc42, when activated, recruits its own GEF Cdc24 (via the scaffold Bem1), thus leading to further Cdc42 activation (**Butty et al., 2002**). Similar positive feedback GEF recruitment could occur in Ras signaling networks. To explore the effects of recruitment-based positive feedback, we made a synthetic GEF in which we fused the C-Raf RBD effector domain to the RasGRF GEF to produce 'RasGRF-RBD'. In this case, the catalytic activity of RasGRF-RBD is always constitutive, but activated Ras will assemble the synthetic GEF on the bead surface to provide a higher effective concentration of GEF and potentially increase the apparent GEF activity (**Figure 7A**). Thus, this feedback mechanism takes an ON GEF molecule and makes it MORE ON as Ras•GTP levels increase. We experimentally measured output responses for this feedback-containing system in 96 system configurations corresponding to four different input strengths, four different GAP concentrations, and six different Ras densities (**Figure 7B**).

The inclusion of recruitment-based positive feedback in the system had considerable consequence for both the dynamics and amplitude of the effector output's we observed. For example, in high GEF / high GAP regimes, multiple local maxima in the output dynamics were observed (**Figure 7—figure supplement 1**). Interesting behaviors were also observed for the output amplitudes. For example, because the fluorescent output-effector and the feedback-effector-GEF compete with one another, signaling amplitude did not increase monotonically with increasing INPUT strength: at the highest level of GEF INPUT will examined (2000 nM), signal was substantially lower than at 1000 nM GEF INPUT, and more comparable in amplitude to the 200 nM GEF INPUT.

How do these differences alter the signaling properties of the system compared to not having this feedback present? Manual inspection of the curves from the system with or without feedback suggested that recruitment-feedback was particularly effective at increasing output in high GAP network contexts. For example, high GEF / low GAP system configurations produced similar responses with or without feedback (**Figure 7C**), but feedback was critical to produce an output from a low GEF / high GAP system configuration (**Figure 7C**). To address this systematically, we calculated a feedback gain for each network configuration we experimentally examined as the ratio of the integrated signal from the feedback system to the non-feedback system. We then interpolated these experimentally determined values for two different Ras density levels to produce phase diagrams showing the impact of feedback on different network configurations (**Figure 7D**).

These data imply that this type of feedback mechanism can produce strong effects in certain network configurations but have little to no effect in other configurations. In particular, networks that contained high amounts of basal GAP-activity but only small amounts of input GEF showed the strongest differences in signal. These correspond to regimes in which the amount of GAP activity in the system dominates the small amount of intrinsic catalytic activity of the GEF, but does not overcome the small amount of localized GEF activity arising from the RasGRF-RBD feedback GEF

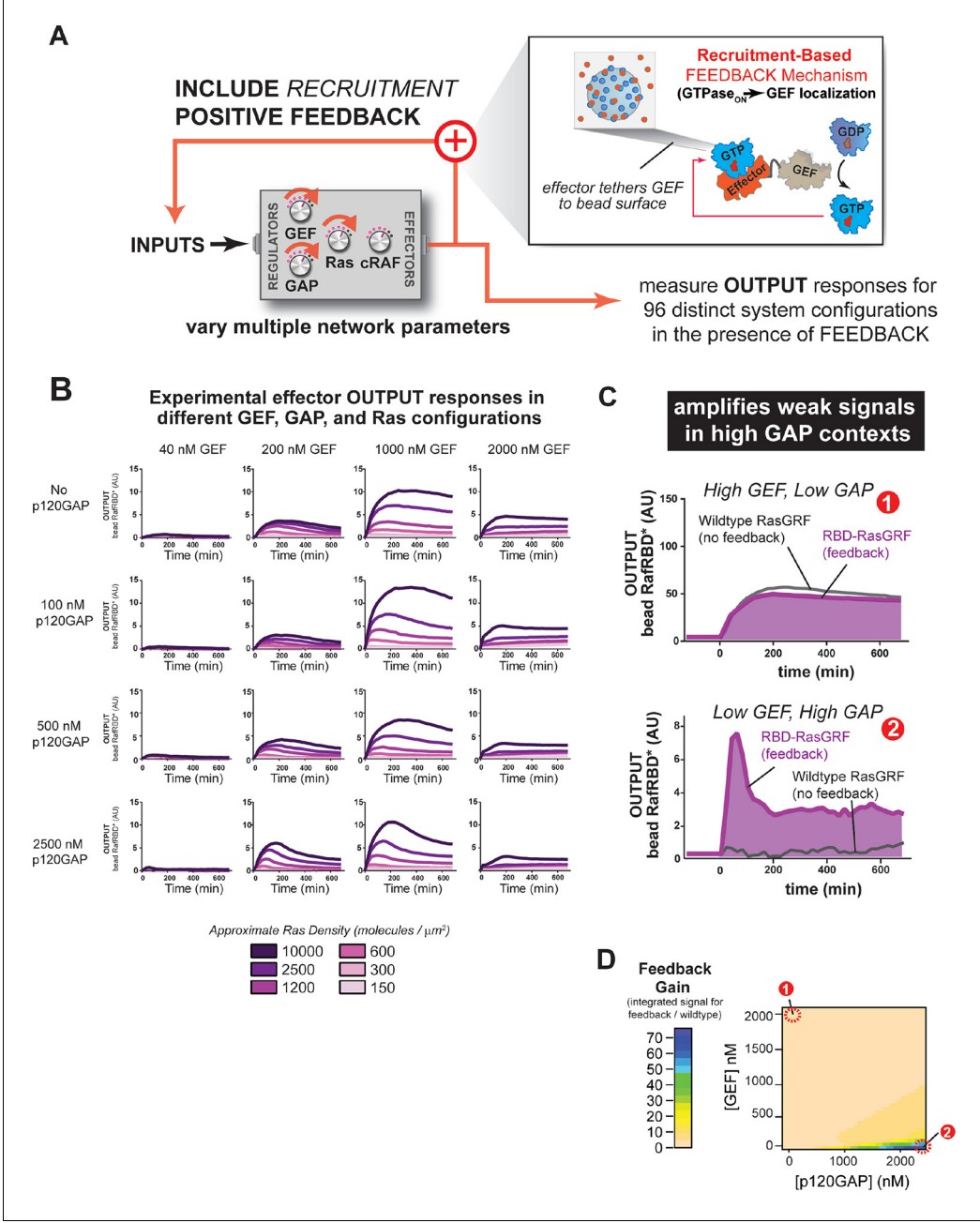

**Figure 7.** Introducing recruitment-based positive feedback into the Ras signaling network alters output dynamics and amplifies weak signals in high-GAP contexts. (**A**) Illustration of Ras system that now includes recruitment-based positive feedback and the synthetic GEF (RasGRF-RBD) that was used to implement the feedback. (**B**) Experimentally determined absolute effector OUTPUT responses across 96 different system configurations. Each graph corresponds to a particular GEF/GAP configuration, and each of the curves within that plot corresponds to a different Ras density as indicated by the color of the curve. (**C**) Examples of output responses for systems under equivalent network configurations (Position 1 of **Figure 7D**) that do (purple line) or do not (grey line) contain recruitment-based feedback. (**D**) Phase diagram depicting the gain provided by recruitment-based feedback (defined as fold-increase in integrated signaling output) in different network configurations. GAP, GTPase-activating protein; GEF, guanine exchange factor; RBD, Ras-binding domain

The following figure supplement is available for figure 7:

**Figure supplement 1.** Normalized (to maximum output) responses of p120GAP/ RasGRF-RBD feedback /RafRBD/ Ras signaling system under a variety of network configurations.

(*Figure 7C*). In contrast, the behaviors of the wild type and feedback systems were most similar under high GEF, low GAP network configurations. In this regime, the GEF activity from the catalytic domain is sufficient to provide strong activation, and any additional boost in activity for localizing the GEF provides only marginal gains (*Figure 7D*). Thus, this type of feedback mechanism seems most powerful for *amplifying signals arising from weak inputs in a high turnover background.*

### Allosteric positive feedback mechanism dampens system overshoot

For the second feedback mode, we replaced the RasGRF GEF with SOScat, the catalytic domain from the Son of Sevenless GEF. SOScat has an intrinsic feedback mechanism built in, such that the molecule has very low GEF activity in the absence of Ras•GTP, but Ras•GTP binding to a distal site on SOScat allosterically stimulates GEF activity to very high levels (*Margarit et al., 2003*; *Freedman, 2006*). Thus, in contrast to the synthetic RasGRF effector fusion (which is always ON), this feedback mechanism takes an OFF GEF molecule and makes it ON in response to Ras•GTP (*Figure 8A*). We experimentally measured output responses for this feedback-containing system in 96 system configurations corresponding to four different input strengths, four different GAP concentrations, and six different Ras densities (*Figure 8B*, normalized responses in *Figure 8—figure supplement 1*).

These data show a very different effect of the allosteric SOScat feedback on signaling outcomes than was observed for the recruitment-based feedback of the RasGRF-RBD fusion. Unlike with recruitment-based feedback, allosteric-feedback did not provide large gains in output amplitudes in any of the regimes we looked like. However, we did observe that transient overshoot behavior was almost completely absent in the output responses of the SOScat system, even in the highest p120GAP condition we inspected. Indeed, the output dynamics from allosteric-feedback networks appeared much more stable and monotonic than those from RasGRF networks that contained no feedback. These effects were most dramatic in high GAP network contexts in which RasGRF produced a large transient overshoot phase, whereas SOScat produced a monotonic approach to the steady state (*Figure 8C*). However, this increased stability was still noticeable even for low GAP high GEF networks in which RasGRF produces a sustained response (*Figure 8C*).

These differences in comparison to the RasGRF-RBD positive feedback system can be at least partially attributed to the OFF→ON feedback mechanism of SOScat in the context of a step-input: for a RasGRF GEF step-input, the system experiences a sudden increase in catalytic activity all at once; for SOScat GEF, although, this step-input in protein levels becomes a *ramp-input* in terms of protein *activity* because it takes multiple rounds of GEF activity to produce enough Ras•GTP for SOScat to achieve significant activity (*Figure 8D*). Thus, this type of feedback mechanism appears to play less of a role in amplifying weak signals and more of a role in dampening dynamic and transient features that would arise naturally in the step-response without this feedback. For SOScat, which mediates Ras activation in response to growth signals, this overshoot minimization might be a desirable feature that reduces the risk of inappropriate proliferation. This use of biochemical feedback is somewhat analogous to the use of feedback control used in some *electronic* systems to avoid overshoot of the steady-state in response to step-inputs.

## Discussion

### One system, many behaviors: a design manual for the diverse signaling behaviors that can be constructed with Ras GTPase systems

In this work, we developed a multi-turnover reconstitution of Ras signaling to explore the space of dynamic output behaviors that could be produced by Ras GTPase systems and to characterize how each network component contributes to these behaviors. Using these assays explored how different perturbations such as oncogenic mutation, component levels, inclusion of additional effector molecules, or introducing positive feedback altered the landscape of available outputs.

Our experiments imply that, much in the same way that a single genome can encode multiple cell types that are regulated through differential gene expression, a single signaling system like Ras can encode multiple dynamic signal processing behaviors by regulating the concentration and identity of network components. This regulation can be *direct* by acting at the level of gene expression. For example, a simple survey of published p120GAP, Ras, and c-Raf mRNA expression levels across a variety of tissue types reveals a staggering amount of diversity in what types of network

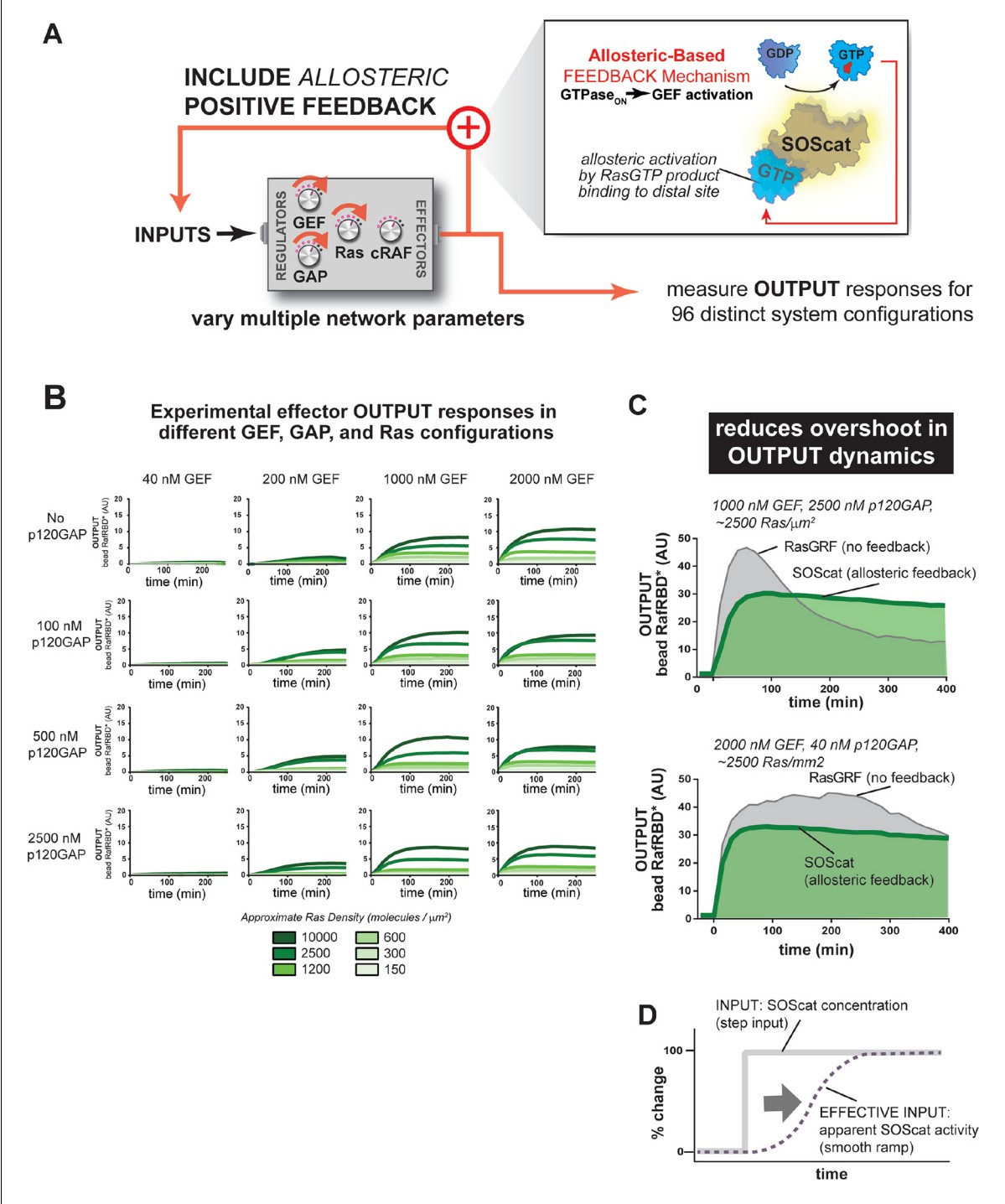

**Figure 8.** Introducing allosteric-based positive feedback into the Ras signaling network reduces transient overshoot and smooths the OUTPUT dynamics. (**A**) Illustration of Ras system that now includes allosteric-based positive feedback and the naturally occurring GEF (SOScat) that was used to implement the feedback. (**B**) Experimentally determined absolute effector OUTPUT responses across 96 different system configurations. Each graph corresponds to a particular GEF/GAP configuration, and each of the curves within that plot corresponds to a different Ras density as indicated by the color of the curve. (**C**) Examples of output responses for systems under equivalent network configurations (high GEF, high GAP) that do (green line) or do not (grey line) contain allosteric feedback. (**D**) Schematic depiction of how an OFF->ON feedback mechanisms converts a step input in SOScat levels into a ramp input in SOScat activity. GAP, GTPase-activating protein; GEF, guanine exchange factor

The following figure supplement is available for figure 8:

*Figure 8 continued on next page*

*Figure 8 continued*

**Figure supplement 1.** Normalized (to maximum output) responses of p120GAP/ SOScat feedback /RafRBD/Ras signaling system under a variety of network configurations.

configurations are present in different cell types and tissues (*Figure 9A*, *Table 1*). The true diversity in these configurations is likely even greater given the plethora of additional GEFs, GAPs, Ras variants and effectors that cells can deploy. Regulating the concentration of these activities can also be achieved *indirectly* by the differential recruitment of these molecules by the receptors that initiate Ras signaling, which changes their *effective concentration* at the plasma membrane. Indeed, many of the catalytic domains that we looked at in this study show regulated interaction dynamics with the plasma membrane in response to extracellular signals (*Gureasko et al., 2010*). Thus, different cells can position their signaling systems at different points in the space of available Ras network configurations and modulate these configurations in response to extracellular cues to provide versatile top-level control of the amplitude and duration of proximal signal processing events (*Figure 9B*).

This versatility is not without trade-offs, however. In particular, we observed many different paths in network-space from one signaling processing behavior to another with much higher or sustained amplitude (*Figure 9C*). These paths include classic oncogenic substitutions like G12V in Ras, but can also be realized by increased GEF activity, decreased GAP activity, or inclusion of high-affinity effectors that increase, extend and sustain signaling responses. While some of these perturbations have not been definitively recognized as drivers of cancer, many are associated with other RASopathies in humans, like Noonan syndrome or type 1 Neurofibromatosis (*Schubbert et al., 2007*; *Bollag et al., 1996*). Thus, the same flexibility that allows Ras systems to realize many different signaling behaviors creates many opportunities for misregulation in response to perturbation.

How any particular perturbation distorted signaling output was highly dependent on network configuration. This was most obvious from comparing the effect of Ras G12V perturbation on GAP-free and high-GAP networks (see *Figure 3*) but is also readily apparent from inspection of the structure of our experimental maps between network configuration and signaling outputs (see *Figure 5—figure supplement 2*). Even networks configurations that produced highly similar output behaviors could nonetheless respond divergently to perturbations. These are network configurations along a contour in the phase diagrams and represent neutral paths along which a cell or species can move without immediate consequence to the system. For example, a low signaling output that is maintained by a weak GEF activity alone might also be produced by a higher GEF activity balanced by a high GAP activity. However, these two configurations would respond very differently to substitution with oncogenic alleles of Ras like G12V. This observation demonstrates the limited predictive power of static steady-state measurements of cellular states and highlights the need to obtain *dynamic* data about the pre-steady state and impulse-response behavior of cellular systems using fine-grained time courses or new methods such as optogenetic pathway activation (*Toettcher et al., 2013*).

The dependence of a perturbation on network configuration can also afford cells new opportunities that may be positive rather than deleterious. As an example, our analysis of the feedback gain produced by the RasGRF-RBD fusion revealed that some configurations had almost no impact on the system output, while others were highly impacted and produced little to no signal without the presence of feedback (see *Figure 7B–D* ). Thus, access to certain regions of this space is not feasible without first acquiring permissive modifications to the feedback architecture of the system. Acquisition of this permissive architecture though, can occur in regions in which there is minimal consequence to the system output. Once this feedback mechanism is present, the space of behaviors available to the system changes and previously non-functional regions of the space can now be accessed.

By interrogating the space of available behaviors to a signaling system in an unbiased way as we have in the present work, we learn not only what the behavior of any particular system configuration is, but also how systems respond to *change* and what paths exist to travel to new configurations with new behaviors. For Ras, this space appears rich with dynamic possibilities and sufficient neutral network structure to provide evolution with ample fodder to facilitate the use of Ras for the wide

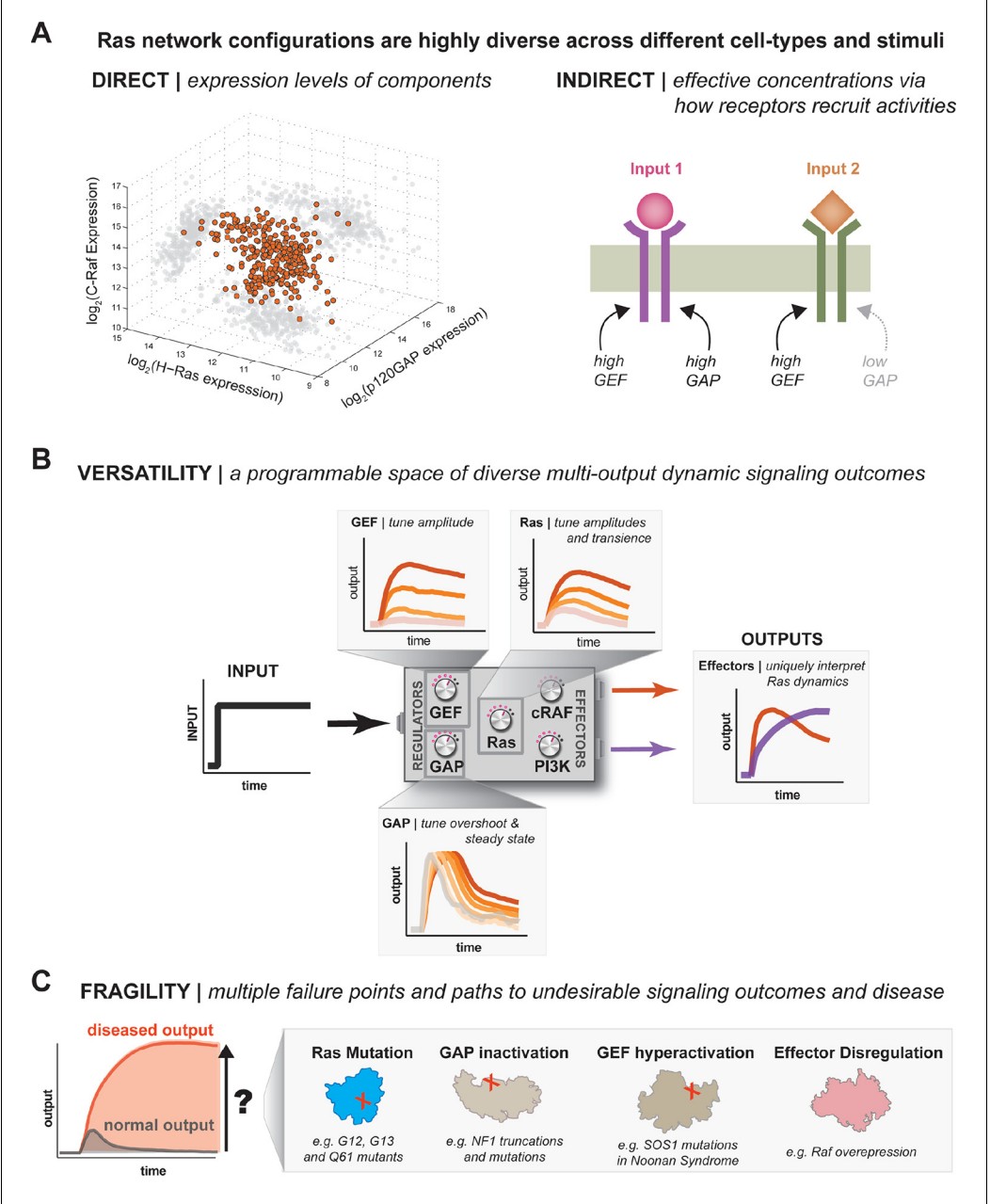

**Figure 9.** One system, many behaviors: versatility and fragility in the space of Ras GTPase signal processing behaviors. (**A**) Illustration of direct and indirect diversity that exists in Ras network configurations. In the direct case, the distribution of p120GAP, H-Ras, and Raf gene expression levels across a variety of human cell types are shown, synthesized from Genevestigator data (see associated *Figure 9–source data 1*). Each orange point corresponds to a cell type and its position in the space indicates its associated expression level in each coordinate. A 'shadow' of each point is projected onto each two-dimensional sub-plane to further clarify the distribution. In the indirect case, a schematic of two receptors that both activate Ras are indicated. One receptor results in strong recruitment of both GEF and GAP, while another only strongly recruits GEF. (**B**) Illustration depicting the versatility of Ras GTPase signaling systems. A simple step-input can be processed into a variety of different dynamic outputs depending on the network configuration. The way in which each network component shapes signaling is illustrated. (**C**) Illustration depicting the fragility of Ras GTPase signaling systems. Given a particular signaling output and a higher level disease output, there exist many paths by which the network configuration can change to produce the diseased output. GAP, GTPase-activating protein; GEF, guanine exchange factor.

The following source data is available for figure 9:

**Source data 1.** Relative gene expression level data from a variety of human tissue and cell types that was used to produce the plot in *Figure 8A*.

**Table 1.** List of plasmids used this study. A description of each construct used in this study, the bacterial antibiotic resistance associated with that plasmid, and a pSC reference index to facilitate any plasmid requests.

| | | Description | Bacteria Marker |
|---|---|---|---|
| pSC | 353 | pMal-H.s.SOS1cat-StrepII | amp |
| pSC | 354 | pMal-H.s.p120GAP(RASA)-StrepII | amp |
| pSC | 369 | pMalStrep-RasGRF(MusGRF1cat ) | amp |
| pSC | 427 | pSNAP-Mal-cRaf-RBD-StrepII | amp |
| pSC | 451 | pSNAP_Mal_H-Ras_2xHis(6xHis-linker-10xHis) | amp |
| pSC | 465 | pMalStrep-RasGRF-30xGAGS-RBD | amp |
| pSC | 485 | pMalStrep-NF1 Ras GAP | amp |
| pSC | 486 | pSNAP-Mal-H-rasG12v-2xHis | amp |
| pSC | 488 | pSNAP-Mal-RafRBD(N64A)-StrepII | amp |
| pSC | 490 | pSNAP-Mal-H-RasG12C-2xHis | amp |
| pSC | 492 | pSNAP-Mal-H-RasQ61L-2xHis | amp |
| pSC | 501 | pSNAP-Mal-ARafRBD-StrepII | amp |
| pSC | 502 | pSNAP-Mal-BRafRBD-StrepII | amp |

array of diverse signaling roles at it plays across different cell types and species, but at the risk of harmful perturbation by diseased alleles or expression states.

## Building distinct signaling output programs by coupling Ras to multiple effectors

One striking observation from this work was the importance of effector molecules in determining how a dynamic Ras•GTP signal is interpreted. This is, in fact, a critical aspect of how these particular signaling systems work as activated Ras itself has no *enzymatic* activity toward other molecules, but instead serves only as a platform for the recruitment of many possible competing effector molecules within the cell. Moreover, activated Ras cannot engage more than one effector simultaneously and thus competition between effectors as well as upstream regulators like GAPs contributes to the system's output dynamics.

For simple one-effector systems that we studied with the C-Raf RBD effector, the concentration of effector shaped not only the amplitude of the output, but also the dynamics of that output as well (see *Figure 4D*). This is because higher effector concentrations equilibrate against their target faster than lower effector concentrations, resulting in different abilities to capture transient features of the target dynamics. Effector concentration is thus more than a passive 'volume' knob that reports on Ras•GTP levels and instead is an active system component that interprets Ras activity to sculpt effector-specific output dynamics owing to its own assembly and disassembly kinetics.

The importance of this property of effectors was even more apparent in two-effector Ras signaling systems, in which we found that equivalent amounts of different competing effectors interpreted the same system inputs with markedly different outputs that differed both in amplitudes as well as in duration and dynamics (see *Figure 6*). These systems showed a variety of interesting multi-effector programs, such as one effector exhibiting a transient response while another was sustained, one effector responding while another did not at all, or two transient responses that peaked and declined with different timing and duration. Moreover, we found that different dynamic behaviors could readily be produced by introducing point mutations in an effector that altered affinity for activated Ras. These observations stress the important roles that both kinetic and thermodynamic aspects of effector assembly and competition play in shaping how an individual effector interprets dynamic Ras•GTP levels in the context of the rest of the network.

An interesting consequence of these different effector behaviors and dynamics is that it can naturally result in the temporal partitioning of distinct activities during a signaling response. This can

allow some effector outputs to be restricted to early phases of signaling, only to decline and be displaced by other more dominant effectors at later stages. These observations extend recent observations of hierarchies of binding by different effectors to Ras under equilibrium conditions with non-hydrolyzable analogs (*Smith and Ikura, 2014*). Thus, the differential perception of Ras•GTP signals by distinct effectors may not be a flaw in the method by which cells make measurements, but a useful feature by which cells can use a single upstream signaling molecule like Ras to dictate a complex temporal program of multiple downstream outputs.

## Additional signal processing mechanisms operate in the context of complex Ras effector binding dynamics

Binding to activated Ras is only the first step in signal propagation for many effectors. For example, binding of Raf kinases we used in this study to Ras primarily serves to deliver the kinase to the plasma membrane where its interaction with lipids (*Ghosh and Bell, 1997*), other Raf kinases (*Freeman et al., 2013*), scaffolds (*Brennan et al., 2011*; *Ritt et al., 2006*) and other macromolecules alters its kinase activity and thus how it sends out signals downstream (REF). In fact, Raf can even activate downstream signaling in the absence of Ras by artificial membrane recruitment (*Stokoe et al., 1994*; *Leevers et al., 1994*). However, this only further emphasizes the importance of effector binding *dynamics* in the context of cellular signal processing, as binding to Ras is a physiological prerequisite for these other mechanisms to take place. Moreover, different Ras effectors such as PI3 Kinase or Ral-GDS will have their own molecule-specific layers of regulation that take place upon interaction with Ras at the plasma membrane. These processes will be influenced by the underlying effector-binding dynamics in different ways depending on the kinetics of these downstream steps. Our work demonstrates that cells have simple systems for modulating and controlling these fundamental binding dynamics and further indicates that known control mechanisms should be analyzed in this complex dynamic context.

Another layer of dynamic and regulatory complexity is also likely to arise as more classes of Ras GTPases are included in the signaling networks. Indeed, our present study has only investigated effector interaction with H-Ras, but there exist many additional Ras isoforms such as K-Ras and N-Ras, K4A-Ras, and K4B-Ras that may engage these effectors in different ways to produce different dynamics. Additionally, there exist related GTPases such as Rap which can serve as platform for Ras effectors but that do not necessarily promote signal propagation (*Wynne et al., 2012*; *Cook et al., 1993*). Understanding how the underlying distribution of GTPase isoforms dictates signal processing behavior is another critical component of cellular signaling and we hope to extend our in vitro system to explore these fundamental questions in the future.

## Systems-level reconstitution as tool to probe the mechanism of biochemical signal processing networks

The network level biochemical approach to interrogating signaling systems we employed in this study occupies a relatively underexplored area in our understanding of cellular decision making systems, but is similar to approaches used to understand dynamic *mechanical* systems in cells like microtubules. Indeed, because additional complexities can emerge when multiple energetically driven processes are coupled together to promote the dynamic assembly and disassembly of competing effectors, exploring how these systems behave in vitro under different configurations sheds new light on the phenomenology of how biochemical signaling devices function and respond to perturbation.

The ability to prepare non-equilibrium steady states in which Ras is actively cycling between ON and OFF states may also prove useful in developing new strategies to ameliorate erroneous signaling associated with diseased states. For example, the fact that we could reconstitute radically different signaling behaviors for wild type and G12V Ras under high GAP conditions is consistent with the notion that wild-type Ras ordinarily cycles quickly but G12V does not. This difference in the *lifetime* of a Ras•GTP molecule compared to RasG12V•GTP could potentially serve as an additional *dynamic* selectivity handle for small molecules in which we want to only target the oncogenic form of Ras. The assays we developed are well suited to compare how small molecules differentially impact signaling through these different forms of Ras side by side under active energy-consuming conditions.

More generally, the simplicity of the approach we present here paves the way for further studies on other types of non-equilibrium signaling systems that center around the assembly of molecules from the cytoplasm on a surface such as the other members of the Ras superfamily like Rho, Rac, Cdc42, and Rab GTPases, as well as other completely different multi-turnover signaling systems like receptor tyrosine kinases or lipid kinases. Some aspects of the H-Ras system may be shared with these systems, while other aspects may be different owing to idiosyncratic features of a particular system of molecules. These systems could also be extended in other ways as well to explore how other biophysical constraints impact these signaling processes. Our reconstitutions could, for example, be extended to lipid-coated beads to explore how membrane fluidity or lipid identity shape effector outputs. Multi-currency networks that include multiple Ras isoforms or contain more than one type of GTPase could also be examined to look at higher-order networks and cascades. Only by building these systems, turning them on and watching them run can we begin to understand how they actually perform and operate in different signaling regimes.

## Materials and methods

### Protein purification

#### Purification, labeling, and nucleotide loading of GTPases

Full-length H-Ras, H-Ras(G12V), H-Ras (G12C), and H-Ras(Q61L) were expressed as N-terminal SNAPtag-(GAGS)$_{2x}$-MBP C-terminal DoubleHisTag (10xHis-(GAGS)$_{3x}$-6xHis) fusion proteins using custom expression plasmids (see plasmids table). The SNAP-tag facilitated labeling with high-performance inorganic dyes for imaging, and the DoubleHisTag on the C-terminus allowed Ni-NTA supports to be loaded stably ($t_{1/2}$ >24 hr) with GTPase in a configuration resembling the native C-terminal attachment mode.

To express protein, BL21(T1R) *E. coli* cells were grown to an OD of 0.4 from a fresh transformation, chilled to 18°C, induced with 0.8 mM IPTG, and allowed to express overnight. The proteins were purified by Ni-NTA affinity chromatography following the manufacturer's instructions but were eluted in the presence of 1M Imidazole to facilitate elution of the DoubleHisTag. Proteins were subsequently purified by amylose affinity chromatography per the manufacturer's guidelines. The protein was then concentrated to ~0.5 mL and purified by gel-filtration chromatography on a Superdex S200 10/300 equilibrated in Nucleotide Exchange Buffer (5 mM EDTA, 20 mM Tris, 150 mM NaCl, pH 8.0) to remove any bound nucleotide from the GTPase.

Labeling of the SNAPtag on the GTPases was performed per the manufacturer's instructions. Briefly, samples were buffer-exchanged into a labeling buffer (150 mM NaCl, 25 mM Tris, pH 7.5) using a zebra desalting column. Following 20-min incubation with 5 mM DTT at 37°C, SNAP-Cell 430 substrate was added at a 1.1:1 dye:protein molar ratio and incubated for 1 hr at 37°C or overnight at 4°C. Unlabeled dye was removed by passing the sample through four zebra desalting columns.

The labeled GTPases were loaded with nucleotide using established protocols (*Eberth and Ahmadian, 2001*). Briefly, GTPases were exchanged into Nucelotide Exchange Buffer using a zebra desalting column and incubated with a 20-fold molar excess of nucleotide (typically GDP) for thirty minutes at room temperature. The loading reaction was quenched by the addition of $MgCl_2$ to 10 mM. Unloaded nucleotide was removed by passing the sample through 4 zebra desalting columns. Samples were finally exchanged into GTPase storage buffer (20 mM Tris, 50 mM NaCl, 10 mM $MgCl_2$, 5% glycerol, pH 7.5), concentrated to ~100 µM, alliquotted, flash-frozen, and stored at -80°C.

#### Purification and labeling of effectors

The RBD effector domains of A-Raf, B-Raf, C-Raf, and any associated mutants were expressed as N-terminal SNAPtag-(GAGS)$_{2x}$-MBP C-terminal Strep-II tag fusions. The SNAP-tag facilitated labeling with high-performance inorganic dyes for imaging, and the Strep-II tag provided a handle for affinity chromatography that did not interact with Ni-NTA supports.

To express protein BL21(T1R) *E. coli* cells were grown to an OD of 0.8 from a fresh transformation, chilled to 18°C, induced with 0.8 mM IPTG, and allowed to express overnight. The proteins were purified by Streptactin affinity chromatography and then amylose affinity chromatography, per the manufacturer's instructions. Proteins were concentrated to ~1 mL and further purified by gel-

filtration chromatography on a Superdex S75 16/60 equilibrated in standard protein buffer (150 mM NaCl, 25 mM Tris pH 7.5).

Labeling of the SNAPtag on the effectors was performed per the manufacturer's instructions. Briefly, samples were incubated with 5 mM DTT at 37°C for 20 min. SNAP-Surface 488 or SNAP-Surface 549 were added at a 1.1:1 dye:protein molar ratio and incubated for 1 hr at 37°C or overnight at 4°C. Unlabeled dye was removed by passing the sample through 4 zebra desalting columns. Samples were concentrated to 100 μM, exchanged into storage buffer (150 mM NaCl, 25 mM Tris pH 7.5, 5% glycerol), aliquotted, flash frozen, and stored at -80°C.

## Purification of GEFs, GAPs, and synthetic effector-GEF fusions
The catalytic domains of RasGRF, p120GAP, NF1-GAP, SOS, and RBD-RasGRF were expressed as N-terminal MBP C-terminal Strep-II tag fusions. The Strep-II tag provided a handle for affinity chromatography that did not interact with Ni-NTA supports.

To express protein BL21(T1R) *E. coli* cells were grown to an OD of 0.4 from a fresh transformation, chilled to 18°C, induced with 0.8 mM IPTG, and allowed to express overnight. The proteins were purified by Streptactin-affinity chromatography and then amylose-affinity chromatography, per the manufacturer's instructions. Proteins were concentrated to ~0.5 mL and purified by gel-filtration chromatography on a Superdex S200 10/300 equilibrated in standard protein storage buffer (150 mM NaCl, 25 mM Tris pH 7.5, 5% glycerol). Proteins were concentrated, aliquotted, flash frozen, and stored at -80°C

## In vitro signal processing assays
### Preparation of GTPase-loaded beads
Here, 50 μL of NiSepharose High-Performance beads (GE Healthcare) were washed twice with 1 mL water, twice with GTPase assay buffer (GAB: 20 mM Tris, 50 mM NaCl, 10 mM $MgCl_2$, 30 mM Imidazole), and resuspended in a final volume of 1 mL GAB. To load the beads, 7.5 μL of this bead slurry was mixed with 7.5 μL of GTPase in a PCR tube and incubated on ice for 1 hr with occasional flicking. The amount of GTPase used to load the beads depending on the desired density for the specific experiments being performed downstream, but a typical bead-loading used 7.5 μL of 30 μM GTPase and typically resulted in Ras densities crudely estimated at 2500 molecules $\times$ $\mu m^{-2}$ (see calculation below). Following incubation, the beads were spun in a table-top minifuge, the supernatant removed, and washed thrice with GAB. The washed beads were resuspended in ~100 μL GAB, transferred to an Eppendorf tube, shielded from light, and stored on ice. The exact amount of final GAB the beads were in was adjusted for any particular experiment such that 2 μL of bead slurry contained roughly 25–100 total beads when placed in a 384-well microscopy plate.

### Preparation of signal processing reactions and data collection
Signal processing reactions were set up in two stages. First, a 'bead-mix' was prepared that contained fluorescent effector at the desired concentration (typically 50 nM) and beads in GAB. It should be noted that the inclusion of 20 mM Imidazole was critical for eliminating non-specific background effector staining on the bead surface and improved reproducibility dramatically. A 20 μL volume of this bead mix was dispensed into the wells of a 384-well Costar microscopy plate. Second, an 'initiation-mix' was prepared that contained fluorescent effector at the desired concentration, 5 mM GTP (or other nucleotide if used) and GEFs and GAPs at the desired concentration, all in GAB. 10 μl of this reaction mix was gently added to the 20 μL bead mix in the 384-well plate to initiate reaction. The large volume of the initiation mix was critical for getting sufficient mixing without the need to pipette up and down and disrupt the beads and improved reproducibility. Once signal processing reactions were initiated, they wells were sealed with PCR plate sealant to prevent evaporation.

All data were collected using a Nikon Eclipse TI inverted microscope equipped with a Yokogawa CSU-X1 spinning disk confocal using a 20x PlanAPO 0.75 NA objective and an electron microscopy charge-coupled device (EM-CCD) camera (Andor, UK). Depending on the experiment, 405 nm, 488 nm, and/or 561 nm wavelength laser light (LMM5, Spectral applied Research) were used for excitation.

For typical experiments, 5–10 x-y positions within a given well were used to collect signal processing behaviors from 20–100 individual beads. Timepoints varied depending on the experiment, but for typical large matrix experiments of 24–60 different GEF/GAP/effector conditions, we typically imaged every 15 min for 6–12 hr. MicroManager software was used to design the imaging protocols and collect the actual data.

## Image analysis, data processing, and statistics

A combination of standard and custom ImageJ macros were used to prepare the primary image data for further analysis. First, drift in the stage was corrected using a macro based around the Multi-StackReg plugin. Two or three color multi-tiff timecourses were split into separate channels. Matrix transformations to register timecourses were obtained using the constant GTPase fluorescence on every bead from the blue channel. These matrices were then used to register the timecourses of the red or green channels. The three channels were then recombined to produce the properly registered multi-tiff used for analysis of the beads. The ImageJ macro code was tweaked depending on the particular experiment, but a representative example of the code is shown below:

```
matrix path= "C:\Users\scoyl_000\Google Drive\MICROSCOPY
\150403\matrix\";
inputpath = "C:\Users\scoyl_000\Google Drive\MICROSCOPY\150403
\raw\";
outputpath = "C:\Users\scoyl_000\Google Drive\MICROSCOPY
\150403\processed\";
function scott_register(input,output,filename)
{
open(input+filename);
run("Split Channels");
run("MultiStackReg", "stack_1=[C3-" + filename + "] action_1=A-
lign file_1=["+matrixpath+"matrix.txt] stack_2=None actio-
n_2=Ignore file_2=[] transformation=[Rigid Body] save");
run("MultiStackReg", "stack_1=[C1-" + filename + "] action_1=
[Load Transformation File] file_1=["+matrixpath+"matrix.txt]
stack_2=None action_2=Ignore file_2=[] transformation=[Rigid
Body]");
run("MultiStackReg", "stack_1=[C2-" + filename + "] action_1=
[Load Transformation File] file_1=["+matrixpath+"matrix.txt]
stack_2=None action_2=Ignore file_2=[] transformation=[Rigid
Body]");
run("Merge Channels...", "c1=C1-" + filename + " c2=C2-" + file-
name + " c3=C3-" + filename + " create");
}
list = getFileList(inputpath);
for (i = 0; i < list.length; i++)
scott_register(inputpath, outputpath, list[i]);
```

We then corrected for uneven sample illumination or background artifacts using a rolling ball background substraction of 50 pixels. These processed images were subsequently used to analyze the signaling behavior of each bead.

To analyze the images, beads were either identified and stored as regions-of-interest (ROIs) automatically from the GTPase fluorescent signal on beads using a custom macro that makes use of ImageJ's 'find particles' function, or in the case of experiments with beads that deliberately contain differing levels of Ras density, beads were identified and stored as ROIs by hand. The particular parameters of the automated bead finding varied depending on the particular experiment, but a representative ImageJ macro is shown below:

```
inputpath = "C:\Users\scoyl_000\Google Drive\MICROSCOPY\150403
\processed\";
```

```
outputpath   =   "C:\Users\scoyl_000\Google   Drive\MICROSCOPY
\150403\beads\";
function scott_findbeads(input,output,filename)
{
open(input+filename);
run("Split Channels");
selectWindow("C3-"+filename);
run("Smooth", "stack");
//set threshold and find particles
Stack.setPosition(1,18,1);
setSlice(20);
setThreshold(3744,23285);
run("Analyze Particles...", "size=400-15000 circularity=0.60-
1.00 display exclude clear include add slice");
//close any and all open windows
close();
close();
close();
//reopen original file
open(input+filename);
//transfer ROIs to overlay and save
run("From ROI Manager");
saveAs("Tiff", output+filename);
//clear manager
roiManager("Delete");
close();
}
list = getFileList(inputpath);
for (i = 0; i < list.length; i++)
scott_findbeads(inputpath, outputpath, list[i]);
```

Once beads were identified and stored as ROIs, a variety of measurements were made for each bead using a custom macro. We measured the average total Ras fluorescence in the ROI, and the average fluorescent effector signal in the ROI at every timepoint in the experiment. We also measured the area and perimeter of the bead. As with other macros, the exact details of the code varied depending on the number of effectors we were simultaneously examining or other aspects of the setup, but a representative ImageJ macro is shown below:

```
inputpath = "C:\Users\scoyl_000\Google Drive\MICROSCOPY\150318
\one_shot_processed\";
outputpath   =   "C:\Users\scoyl_000\Google   Drive\MICROSCOPY
\150318\one_shot_data\";
setBatchMode(true);
list = getFileList(inputpath);
for (i = 0; i < list.length; i++)
crunchimage(inputpath, outputpath, list[i]);
setBatchMode(false);
//The Functions that are used in the macro are below
function crunchimage(input,output,filename)
{
//open the image
open(input+filename);
//clear log file and reulsts table
run("Clear Results");
print("\Clear");
//import ROIs from overlay
```

```
run("To ROI Manager");
//count ROis
count=roiManager("count");
//record Ras AMPS and RBD timecourse for each ROI
for(i=0;i<count;i++)
{
// first record the AMP for the Ras field;
recordAMP(i);
// next calculate the time series for the data
recordTimecourse(i);
// print newline marker
print("!");
}
selectWindow("Log");
saveAs("Text", output+filename);
close();
}
function recordAMP(index)
{
//function that reports area, mean intensity, and perimieter
// of an ROI used to get everything EXCEPT the timecourse data
run("Clear Results");
roiManager("Select",index);
// select channel 2 and select the midpoint of hte stack
Stack.setPosition(2, 18,18);
// make measurements
run("Measure");BeadArea=getResult("Area",0);BeadMean=getRe-
sult("Mean",0);BeadPerim=getResult("Perim.",0);print(Bea-
dArea+","+BeadMean+","+BeadPerim+",");
}
function recordTimecourse(index)
{
run("Clear Results");
roiManager("Select",index);
sliceCount=nSlices()/2;
// print(sliceCount);
for(k=0;k<sliceCount;k++)
{
Stack.setPosition(1, 18,k+1);
run("Measure");
timepointK=getResult("Mean",k);
print(timepointK+",");
}
}
```

The output of this macro is a file that contains a list of every single bead trace in the multi-tiff image. Each trace begins with the area measurement of the bead, the mean total Ras intensity of the bead, and the perimeter measurement of the bead, followed by the effector measurement of the bead at every timepoint. Each measurement ends with a ',' and is on a newline. At the end of each bead trace, a stop marker '!' is printed. These data are then transformed by GREP and shell script into a CSV where each line contains all the relevant information about each single bead's signaling trace. These data can then be loaded into either Matlab or Excel for further analysis.

Once in Excel, data were typically further analyzed as follows: (i) intensity measurements were normalized to perimeter instead of area, (ii) the time-series data for a given bead was normalized such that the time-zero effector measurement was zero, and (iii) single bead traces were binned based on total GTPase levels to obtain statistics on the signaling behavior. Beads were assigned to

the nearest of the 6 Ras density beads: 150 molecules / μm², 300 molecules / μm², 600 molecules / μm², 1200 molecules / μm², 2500 molecules / μm², 10,000 molecules / μm². The individual bead traces within a given bin were then averaged together to produce an average response for the associated density bin and network configuration. Each trace was typically the average of 15–80 beads from the combination of two independent experiments. The standard error of the mean for a given trace was typically <15%.

## Estimation of Ras density

We estimated the approximate Ras density of a bead in molecules $\times$ μm$^{-2}$ in the following manner. First, we determined the correspondence between the concentration of labeled Ras and its fluorescence intensity by imaging serial dilutions of known concentrations of SNAP-Cell 430 labeled Ras in solution. From this, we could associate a particular fluorescence intensity with a three-dimensional concentration in μM. For our imaging conditions, this relationships was:

Concentration (μM) = 4.287 (μM $\times$ AU$^{-1}$) $\times$ Intensity (AU)

For any individual bead then, there is some maximum fluorescence intensity on the bead surface that we can associate with an apparent three-dimensional concentration. This apparent three-dimensional concentration can be used to estimate an inferred two-dimensional density by assuming that the apparent three-dimensional concentration is a consequence of molecules within some two-dimensional area exploring three-dimensional space as constrained by some confinement length $h$ (*Zhang et al., 2006*; *Wu et al., 2011*). We use an $h$ obtained from previously published work on EGFR kinases tethered by His-tags to a DGS-NTA(Ni) charged vesicle surface, which assumed a radius of confinement of 55 nm (*Zhang et al., 2006*). Given that 1 μM corresponds to ~602 molecules / μm³, this enabled us to convert between three-dimensional and two-dimensional concentrations using the relation:

1 μM $\times$ (602 [molecules $\times$ μm$^{-3}$] $\times$ μM$^{-1}$) * 0.055 μm = 33.1 molecules μm$^{-2}$
$\Rightarrow$ 2-D Concentration (molecules $\times$ μm$^{-2}$) =
3-D concentration (μM) * 33.1 ([molecules $\times$ μm$^{2}$] x μM$^{-1}$)

We stress that this is only an *estimate* of the Ras density and should not be taken as a highly accurate assessment of the Ras density. Nonetheless, it provides a crude estimation that indicates that our experiments are not operating in a highly non-physiologic regime. Importantly, for our analysis the exact number of Ras molecules on the bead surface is not critical. Indeed, the relative abundances of Ras on different beads is more important as it enables us to compare behaviors between beads as Ras densities change.

## Kinetic modeling and simulation

Kintek Student Explorer (*Johnson et al., 2009*) was used to simulate the dynamic behavior of a variety of models for GTPase activation. Time was modeled in seconds and concentrations in nanomoles. Rate constants for association and dissociation of molecules from the GTPase were based on published Biacore measurements (*Fischer et al., 2007*), and catalytic rate constants for GEF and GAP activities were based published solution measurements (*Bollag and McCormick, 1991*; *Freedman, 2006*). GEF was not modeled explicitly but rather directly incorporated in the rate constant for nucleotide release. Each simulation was allowed to run for 42000 s (~700 min).

Three models were initially explored. The first model was a two-state GTPase model that did not account for competition between GAP and effector. This was modeled in Kintek using the following equations and parameters:

|  | k- | k+ |
|---|---|---|
| G + T = GT | 1 | 0 |
| GT = GD | 0.0001*[GAP] | 0 |
| GD = G + D | 0.05 | 0 |
| GT + EFF = GT_EFF | 0.0001 | 0.001 |
| GT_EFF = GD + EFF | 0.0001 | 0 |

The second model was a two-state GTPase model that explicitly modeled competition between GAP and effector. This was modeled in Kintek using the following equations and (physiological) parameters:

| | k+ | k- |
|---|---|---|
| G + T = GT | 1 | 0 |
| GT = GD | 0.0001 | 0 |
| GD = G + D | 0.00505 | 0 |
| GT + EFF = GT_EFF | 0.0001 | 0.001 |
| GT_EFF = GD + EFF | 0.0001 | 0 |
| GT + GAP = GT_GAP | 0.0001 | 0.01 |
| GT_GAP = GD + GAP | 1 | 0 |

The third model was a three-state GTPase model that included an additional post-hydrolysis GTPase state (GI) which was refractory to GEF activation. This state converts to the GDP form on a slow timescale. This was modeled in Kintek as:

| | k+ | k- |
|---|---|---|
| G + T = GT | 1 | 0 |
| GT = GD | 0.0001 | 0 |
| GI = GD | 0.0001 | 0 |
| GD = G + D | 0.005 | 0 |
| GT + EFF = GT_EFF | 0.0001 | 0.001 |
| GT_EFF = GD + EFF | 0.0001 | 0 |
| GT + GAP = GT_GAP | 0.0001 | 0.01 |
| GT_GAP = GD + GAP | 1 | 0 |

The third model was the best at explaining the transient behaviors of the system that we observed as well as the differences between WT and G12V Ras, and thus was used as the basis of all subsequent modeling and simulations. For any given simulation in the text, the initial conditions and any changes to associated rate-constants are indicated in the figure legend.

## Comparison of network configurations across cell and tissue types

Relative log-transformed expression levels for p120GAP, C-Raf, and H-Ras across a variety of cell-types and tissue-types were obtained from data contained within the Genevestigator software package (see data in *Table 1*). The three-dimensional phenotypes associated with each cell or tissue type was plotted as a 3D scatterplot using Matlab.

## Acknowledgements

SMC was supported by a National Science Foundation Graduate Research Fellowship. This work was supported by NIH grants RO1 GM55040, PN2 EY016546, P50 6M081879 (WAL), and the Howard Hughes Medical Institute (WAL). We thank T Anooki, L Bugaj, R Gordley, RS Isaac, A Mitchell, G O' Donoghue, K Roybal, M Thomson, J Walter, and A Weeks for helpful discussions.

## Additional information

### Funding

| Funder | Grant reference number | Author |
|---|---|---|
| National Institutes of Health | RO1 GM55040 | Wendell A Lim |
| National Institutes of Health | PN2 EY016546 | Wendell A Lim |
| National Institutes of Health | P50 6M081879 | Wendell A Lim |
| Howard Hughes Medical Institute | | Wendell A Lim |
| National Science Foundation | Graduate Research Fellowship | Scott M Coyle |

The funders had no role in study design, data collection and interpretation, or the decision to submit the work for publication.

### Author contributions

SMC, Conception and design, Acquisition of data, Analysis and interpretation of data, Drafting or revising the article; WAL, Conception and design, Analysis and interpretation of data, Drafting or revising the article

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
