## [Decision Letter]

Thank you for submitting your work entitled "Mapping functional versatility and fragility of Ras GTPase signaling circuits through in vitro network reconstitution" for consideration by *eLife*. Your article has been reviewed by three peer reviewers, and the evaluation has been overseen by a Reviewing Editor (Phil Cole) and Tony Hunter as the Senior Editor.

The reviewers (Mark Philips, Mike Rosen, and Shu-Ou Shan) have discussed the reviews with one another and the Reviewing Editor has drafted this decision to help you prepare a revised submission.

Summary:

The authors present an ingenious study of signal processing down the Ras pathway. These investigators develop an exceedingly simple yet powerful system: measure the recruitment in vitro to Ras-coated microbeads of fluorescently tagged Ras binding domains (RBDs) of various effectors to quantify the amplitude and kinetics of Ras signaling. By manipulating the density, mutational or GTP-binding state of the bound Ras along with the concentrations of single or multiple GEFs, GAPs and effectors, the investigators are able to observe a wide variety of signal outputs. Among the most compelling results is the dependence on GAP concentrations of the difference in signal output between wt and activated Ras (Figure 3). This type of in vitro systems approach to Ras signaling provides much fodder for discussion and future investigation and contributes to our understanding of how a single pathway with a small number of nodes can generate a wide-variety of outputs. In addition, the manuscript is clearly written and the graphically didactic and uniform figures help make an inherently dense set of data digestible. Accordingly, the manuscript is suitable for publication in *eLife*. However, it is recommended that several issues be addressed.

Essential revisions:

1) The manuscript would be deeper and ultimately more satisfying if the phenomena could be explained in terms of the biochemical properties of the component species (rate constants, binding affinities, etc.). It might be said that the majority of observations from Figure 1–Figure 6 can be predicted from a simple molecular model using only four free parameters: (i) the rate constant of GDP -> GTP exchange to generate GTP-bound Ras; (ii) the rate constant of GTP hydrolysis to generate GDP-bound Ras; (iii) and (iv), the rate constants for effector binding to and dissociating from GTP-bound Ras. Different responses were observed because each of these parameters can be changed by modulating concentration, introducing allostery or competition. Furthermore, this modeling should take into account predicted cellular concentrations of the various components. Constructing such a model and comparing the predictions (a simple kinetic simulation) with the empirical data observed here will significantly enhance the value of this work, and any deviations from the prediction will suggest new mechanisms or biology. Most of the explanations in the text for individual pieces of data will likely also be folded into this more fundamental, organizing model.

2) While no new experiments are requested on this point, discussion is needed on how the absence of membranes in this system may alter the interpretation of the results. Ras proteins are peripheral membrane proteins that, although they appear to stably associate with membranes at steady state, are, in fact, in a dynamic equilibrium both in terms of association with the membrane per se (see PMID 16236799) and within membrane microdomains (see PMID: 17618274). The behavior of the stationary Ras proteins affixed to the inert microbeads is likely quite different from their membrane-asociated counterparts in an in vivo system. The lack of membrane also confounds other aspects of interpreting the results. For example, in their analysis the authors equate recruitment of the RBD of an effector to the microbead as a readout of signaling. But this is simplistic. Raf-1 is a kinase that is the first node in a cascade of kinases. The sine qua non of Raf-1 signaling involves stimulation of the kinase activity of Raf-1. Yet the interaction of Raf-1 with GTP-loaded Ras does not stimulate kinase activity in the absence of a cellular membrane (PMID: 8622647, 7811320). Indeed, the membrane acts upon the Raf-1 kinase in a manner that involves the cysteine-rich domain and initiates a poorly understood activation sequence, which may include dimerization. Importantly, Raf-1 sent to the membrane via a CAAX sequence does not require GTP•Ras at all for full activation (PMID: 8196769, 7811320) demonstrating that it is membrane not Ras that activates the kinase. Likewise GEFs for Ras and other small GTPases require biological membranes for activation. Indeed, the way GEFs like SOS and RasGRPs are regulated is by modulation of their affinity for membranes utilizing domains like SH2 (on Grb2), PH and C1 for membrane binding. Although the catalytic domain of SOS will have exchange activity toward Ras in solution, in vivo the histone, PH and DH domains are required for productive associate with membranes (PMID: 20133692).

3) We could not find information about sample size and statistical analysis of data. This needs to be laid out clearly.

---

## [Author Response]

*The manuscript would be deeper and ultimately more satisfying if the phenomena could be explained in terms of the biochemical properties of the component species (rate constants, binding affinities, etc.). It might be said that the majority of observations from Figure 1–Figure 6 can be predicted from a simple molecular model using only four free parameters: (i) the rate constant of GDP -> GTP exchange to generate GTP-bound Ras; (ii) the rate constant of GTP hydrolysis to generate GDP-bound Ras; (iii) and (iv), the rate constants for effector binding to and dissociating from GTP-bound Ras. Different responses were observed because each of these parameters can be changed by modulating concentration, introducing allostery or competition. Furthermore, this modeling should take into account predicted cellular concentrations of the various components. Constructing such a model and comparing the predictions (a simple kinetic simulation) with the empirical data observed here will significantly enhance the value of this work, and any deviations from the prediction will suggest new mechanisms or biology. Most of the explanations in the text for individual pieces of data will likely also be folded into this more fundamental, organizing model.*

This was a very informative suggestion. We have now explored a number of kinetic models and included figures about their associated simulations in the manuscript. An important result from this modeling was the observation that the simplest model (as was suggested by the reviewers above) was unable to produce the types of transient overshoot behaviors that we observed in our data. Indeed, the fact that we could not produce overshoot from the simplest model is consistent with theoretical work on n-state Markov systems, which argue that this type of behavior is only possible for systems with at least three states. We found that overshoot could be introduced in the system by either (i) including competition between GAP and effector for the GTPase and/or (ii) including a post-hydrolysis intermediate GTPase state that is refractory to nucleotide reloading. While (i) could formally produce overshoot behavior, it could only do so in non-physiologic parameter regimes for GAP and was unable to explain the lack of overshoot in G12V compared to WT (the model predicted overshoot in both scenarios). In contrast, including both competition *and* a refractory intermediate GTPase state produced a model that captured the main observations as to how system output was affected by GAP, Ras density, effector concentrations, mutational state and multiple effectors. The exact molecular nature of this intermediate GTPase state will be the subject of additional work but at present is outside the scope of this systems-oriented manuscript. We have included an extensive discussion of this modeling in the main text, and have included 3 new associated supplemental figures that contain this modeling data and articulate how it connects to our empirical observations.

*While no new experiments are requested on this point, discussion is needed on how the absence of membranes in this system may alter the interpretation of the results. Ras proteins are peripheral membrane proteins that, although they appear to stably associate with membranes at steady state, are, in fact, in a dynamic equilibrium both in terms of association with the membrane per se (see PMID 16236799) and within membrane microdomains (see PMID: 17618274). The behavior of the stationary Ras proteins affixed to the inert microbeads is likely quite different from their membrane-asociated counterparts in an in vivo system. The lack of membrane also confounds other aspects of interpreting the results. For example, in their analysis the authors equate recruitment of the RBD of an effector to the microbead as a readout of signaling. But this is simplistic. Raf-1 is a kinase that is the first node in a cascade of kinases. The sine qua non of Raf-1 signaling involves stimulation of the kinase activity of Raf-1. Yet the interaction of Raf-1 with GTP-loaded Ras does not stimulate kinase activity in the absence of a cellular membrane (PMID: 8622647, 7811320). Indeed, the membrane acts upon the Raf-1 kinase in a manner that involves the cysteine-rich domain and initiates a poorly understood activation sequence, which may include dimerization. Importantly, Raf-1 sent to the membrane via a CAAX sequence does not require GTP•Ras at all for full activation (PMID: 8196769, 7811320) demonstrating that it is membrane not Ras that activates the kinase. Likewise GEFs for Ras and other small GTPases require biological membranes for activation. Indeed, the way GEFs like SOS and RasGRPs are regulated is by modulation of their affinity for membranes utilizing domains like SH2 (on Grb2), PH and C1 for membrane binding. Although the catalytic domain of SOS will have exchange activity toward Ras in solution, in vivo the histone, PH and DH domains are required for productive associate with membranes (PMID: 20133692).*

This is an outstanding point and clarification of what we are measuring (binding) and how that is reflected in ‘signaling outputs’ has been incorporated into the manuscript. We emphasize that it was not our intent to suggest that other critical components of Raf activation (allostery, dimerization, lipid interactions, etc.) were not important in signal processing. Instead, we were primarily focused in how effector binding to GTPases (the most fundamental aspect of all Ras-mediated signal transmission) *in general* is affected by network configurations and effector properties. The complex effector binding dynamics that we observe are now discussed as the background and context in which other molecule-specific signaling mechanisms like dimerization or lipid interactions take place. Extending our in vitro reconstitution to lipid supports and post-binding readouts is something we are currently exploring in ongoing work and we look forward to seeing how molecule-specific phenomenon interface with the complex binding dynamics we observed in this work.

We could not find information about sample size and statistical analysis of data. This needs to be laid out clearly.

We have extended our methods discussion to clearly indicate this information.